# Co-expression analysis reveals distinct alliances around two carbon fixation pathways in hydrothermal vent symbionts

**Jessica H. Mitchell** ✉, **Adam H. Freedman**, **Jennifer A. Delaney** & **Peter R. Girguis** ✉

Most autotrophic organisms possess a single carbon fixation pathway. The chemoautotrophic symbionts of the hydrothermal vent tubeworm *Riftia pachyptila*, however, possess two functional pathways: the Calvin–Benson–Bassham (CBB) and the reductive tricarboxylic acid (rTCA) cycles. How these two pathways are coordinated is unknown. Here we measured net carbon fixation rates, transcriptional/metabolic responses and transcriptional co-expression patterns of *Riftia pachyptila* endosymbionts by incubating tubeworms collected from the East Pacific Rise at environmental pressures, temperature and geochemistry. Results showed that rTCA and CBB transcriptional patterns varied in response to different geochemical regimes and that each pathway is allied to specific metabolic processes; the rTCA is allied to hydrogenases and dissimilatory nitrate reduction, whereas the CBB is allied to sulfide oxidation and assimilatory nitrate reduction, suggesting distinctive yet complementary roles in metabolic function. Furthermore, our network analysis implicates the rTCA and a group 1e hydrogenase as key players in the physiological response to limitation of sulfide and oxygen. Net carbon fixation rates were also exemplary, and accordingly, we propose that co-activity of CBB and rTCA may be an adaptation for maintaining high carbon fixation rates, conferring a fitness advantage in dynamic vent environments.

Carbon fixation provides the organic carbon found in all biomass. Six different carbon fixation pathways have been characterized[1] and more have been proposed[2–4]. Nearly all autotrophic species possess just one known carbon fixation pathway, which typically reflects their evolutionary history and the reduction/oxidation state of their environment[5,6]. Some organisms use carbon fixation machinery for other cellular functions, for example, maintaining intracellular redox balance during fermentation[5], while others possess multiple enzyme isoforms with varying catalytic properties that are thought to expand their ecological niche[6]. More recently, two carbon fixation pathways were found in the chemoautotrophic endosymbionts of hydrothermal vent tubeworms and closely related free-living bacteria:

the Calvin–Benson–Bassham (CBB) and the reverse tricarboxylic acid (rTCA) cycles, both expressed and active[7,8].

However, we have a limited understanding of (1) whether these pathways are constitutively active and (2) how this activity relates to environmental conditions, and (3) have no understanding of how these pathways integrate with other metabolic processes.

Here we studied the vent tubeworm *Riftia pachyptila* (hereafter called *Riftia*) and its endosymbiotic bacteria ('*Candidatus* Endoriftia persephone', hereafter called Endoriftia) to investigate how the external environment influences the expression and activity of symbiont CBB and rTCA. Our goals were to (1) discern the relationship between gene/pathway expression and environment, (2) elucidate how these

Harvard University, Cambridge, MA, USA. ✉e-mail: jessicamitchell@fas.harvard.edu; pgirguis@oeb.harvard.edu

**Fig. 1 | Background, study design and methods used. a**, The hydrothermal vent habitat of the East Pacific Rise, the ecological niche of the *Riftia* symbiosis. Image courtesy of Schmidt Ocean Institute. **b**, Diagram of the location of Endoriftia within the *Riftia* host. **c**, Simplified illustration of Endoriftia's substrate utilization. **d**, High-pressure respirometry system, an onboard tool for facilitating experiments under in situ-like conditions. **e**, The conditions that were varied in experiments done on live *Riftia*. **f**, Post-experimental evaluation of *Riftia* tissue and symbiont-containing tissues: ${}^{13}C$ carbon incorporation, gene expression and co-expression analysis.

pathways interact with other metabolic processes, and (3) robustly measure the rates of carbon fixation and incorporation. Remarkably, *Riftia* is one of the fastest growing marine invertebrates and can achieve biomass densities comparable to tropical forests[9–11]. *Riftia* resides in diffuse hydrothermal flows where hydrothermal fluid containing hydrogen sulfide ($\Sigma H_2S$: sum of sulfide species $H_2S$, $HS^-$ and $S^{2-}$), hydrogen ($H_2$) and elevated dissolved inorganic carbon ($\Sigma DIC$: dissolved species of inorganic carbon) mixes with seawater replete with oxygen ($O_2$) and nitrate ($NO_3^-$) (Fig. 1a). This environment is characterized by extreme and rapid shifts in temperature and geochemistry. The symbionts of *Riftia* oxidize $\Sigma H_2S$ with $O_2$ (and to an extent $NO_3^-$)[12] to fix carbon, serving as the sole organic carbon source for both symbiont and worm[13]. The symbiont also converts nitrate into bioavailable nitrogen[12]. *Riftia* brings the necessary substrates for these processes from the vent environment to the symbionts via well-developed vasculature, haemoglobin and ion transporters[14] (Fig. 1b,c).

This organism's high productivity may be due to the activity of two carbon fixation pathways, but so far there are no data that relate transcript and protein expression to metabolic rate. Also, Endoriftia's CBB and rTCA pathways are non-canonical. The CBB cycle lacks two essential enzymes that are likely replaced by a pyrophosphate-dependent 6-phosphofructokinase (PPi-PFK)[15], potentially increasing efficiency by up to 30%[16]. The rTCA genes of Endoriftia are clustered with genes that encode enzymes involved in electron transfers: a NADH dehydrogenase/heterodisulfide reductase (Hdr-Flx) complex potentially involved in flavin-based electron bifurcation (FBEB), an $H^+$ translocating $NAD(P)^+$ transhydrogenase (encoded by pntAB) and an $Na^+$ Rnf membrane

complex (hereafter, Rnf)[8]. This clustering suggests energy conservation via FBEB within the rTCA, which could shift the thermodynamic constraints of this pathway. However, while it is tempting to assert that the rTCA pathway has advantages over CBB because it generally requires less ATP per unit carbon fixed, differences in their mechanisms and the oxygen sensitivity of rTCA due to its dependence on reduced ferredoxins ($Fd_{red}$) complicate direct comparisons[17,18].

Here we present the results from a series of experiments with live *Riftia* in our high-pressure respirometric system. We exposed 30 individuals to a range of environmentally relevant geochemical conditions, with in situ-like temperature and pressures. We measured dissolved $\Sigma H_2S$ and $O_2$ uptake during incubations, directly quantified carbon fixation and incorporation rates using ${}^{13}C$-labelled inorganic carbon and sampled individual worms post treatment for metatranscriptomic sequencing to study patterns of differential gene expression (DE). From the latter, we built a co-expression network that provides an in-depth look at an organism that operates two carbon fixation pathways (Fig. 1d–f). We observed patterns of co-expression that reveal both CBB and rTCA as metabolic hubs, with alliances (significant co-expression) to other distinct metabolic processes. Rates of carbon incorporation were also among the highest for autotrophs in natural environments. When seawater $\Sigma H_2S$ and/or $O_2$ were limited, co-expression analyses implicated the upregulation of the rTCA and a [NiFe] hydrogenase as being among the most significant of metabolic responses. Finally, the distinct bi-modal distribution of other metabolic processes around two functionally degenerate carbon fixation pathways underscores the importance of both and hints

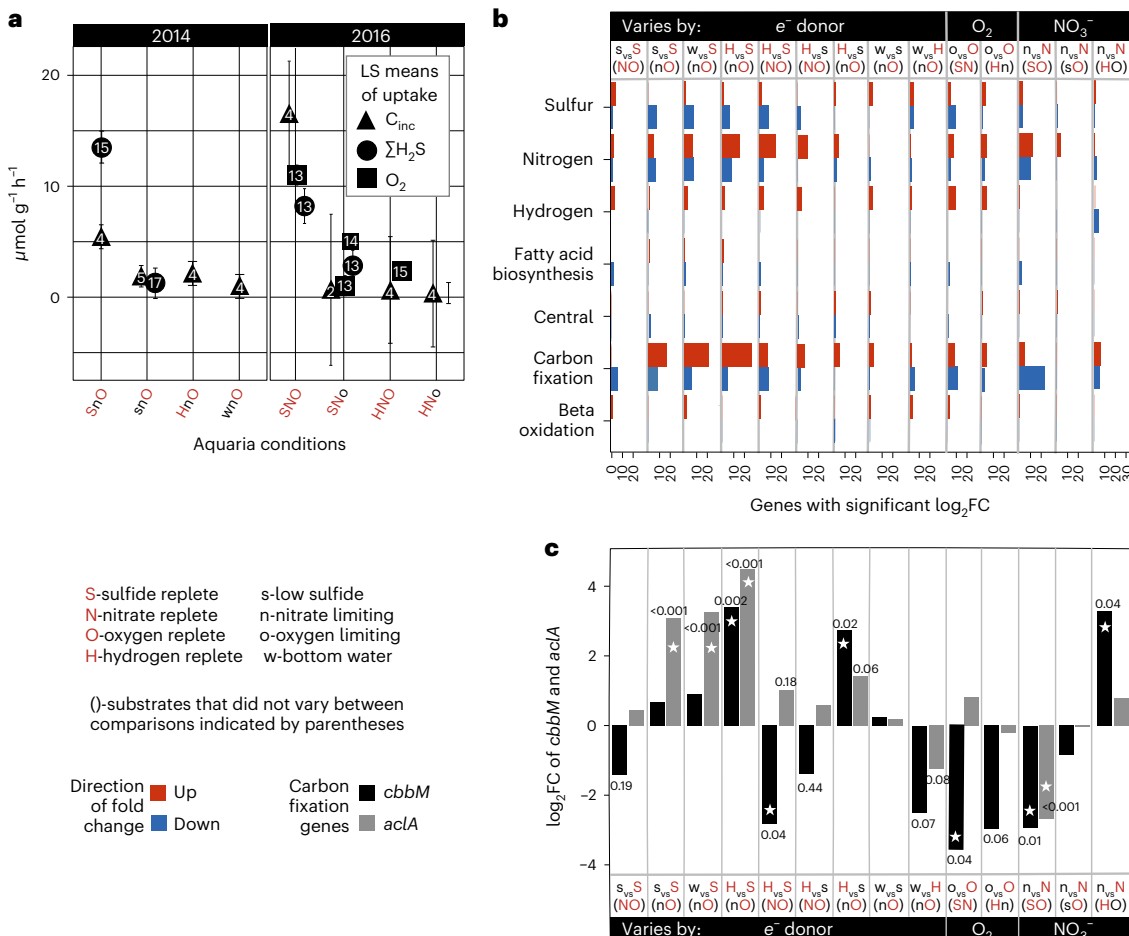

**Fig. 2 | Uptake rates and gene expression under varying conditions.**
**a**, Symbols represent different measurements: triangles for $C_{inc}$ in symbiont-containing tissues, circles for $\Sigma H_2S$ and squares for $O_2$ uptake rates ($O_2$ not measured in 2014). Inside the shapes, numbers indicate the count of independent samples: individual worms for $C_{inc}$ and separate sampling events for $\Sigma H_2S$ and $O_2$. Error bars indicate the standard errors of the least square means. See Suppmentary Table 1 for details on aquaria conditions. **b**, Number of genes that had significant DE for each metabolic category. The $\log_2$-transformed fold change ($\log_2FC$) calculated using the limma package in R with a two-sided linear model fit, followed by empirical Bayes moderation of the standard errors (using the eBayes function with the 'robust=TRUE' parameter) with a $P_{adj} \leq 0.05$ (using the Benjamini–Hochberg

method for FDR, controlling for multiple comparisons) and a $\log_2FC \geq |{-1}|$. Red bars indicate the number of genes that showed a relative increase in expression, blue bars indicate the number of genes that showed a relative decrease in the two treatments being compared. Comparison of aquaria conditions represented by letters on top row, such that the first letter is the condition that indicates a log fold change (in comparison to the condition represented by the second letter). Letters in parentheses indicate the substrates that did not vary between comparisons. **c**, $\log_2FC$ in *cbbM* and *aclA* (calculated as described above), representative genes for the CBB and the rTCA cycles, respectively. Star indicates $P_{adj} < 0.05$, genes that showed significant DE. Aquaria treatment comparisons are represented by letters at the bottom (see abbreviation key).

at a regulatory independence that may offer more protection from environmental perturbations[19].

## Results

### *Riftia* substrate uptake rates during experimental treatments

The $\Sigma H_2S$ and $O_2$ uptake rates of *Riftia* during treatments were highly correlated (Fig. 2a and Supplementary Table 2), which is consistent with previous high-pressure respirometric studies[12,20,21]. By contrast, $H_2$ uptake rates during treatments were not significantly different from those in control aquaria and did not correlate with $O_2$ uptake[22] (Fig. 2a). Results from statistical tests comparing substrate uptake rates among treatments are reported in Supplementary Table 2.

### *Riftia* carbon incorporation

*Riftia* subjected to experimental treatments showed substantial incorporation of the amended $^{13}C$-bicarbonate, yielding stable carbon isotope ratios far heavier than the observed natural abundance ($n = 37$; Supplementary Table 1), consistent with an organism that

requires both sulfide and oxygen for autotrophic growth. All experimental conditions and rates of $^{13}C$-labelled inorganic carbon incorporation rates ($C_{inc}$) are summarized in Supplementary Table 1. Results of statistical tests comparing $C_{inc}$ rates among aquaria are reported in Supplementary Table 2.

### Overall patterns of symbiont gene expression

Of the protein-coding genes, 4.4–15.7% were central carbon genes that are part of either the rTCA or the CBB pathways (many of which are also found in the TCA, pentose phosphate pathways (PPP), glycolysis and gluconeogenesis).

Extended Data Fig. 1 shows 14 pairwise comparisons, with significant DE being observed in up to 831 genes. The log-transformed fold change (logFC) versus log-transformed counts per million (CPM) of these comparisons show a wide range of responses, with the highest DE seen in treatments comparing $\Sigma H_2S$ to $H_2$ (with replete $O_2$), as well as in treatments comparing replete $NO_3^-$ to the absence of dissolved $NO_3^-$ (in these cases, $\Sigma H_2S$ and $O_2$ were both replete).

### DE among functional groups

Genes related to energy (C) and signal transduction (T) exhibited higher DE in treatments with limited or no dissolved $\sum H_2S$ and/or $O_2$, compared with genes involved in translation, ribosome structure and biogenesis (J), as well as inorganic ion transport (P) (Extended Data Fig. 2). The energy (C) category was further sorted: carbon fixation, beta oxidation, nitrogen metabolism, sulfur metabolism, fatty acid biosynthesis and hydrogenases. A comparison of DE among these groups highlighted an increase in DE of carbon fixation genes (some bidirectional, hence could indicate oxidative or anaplerotic reactions), hydrogenases and genes related to nitrogen metabolism in many $\sum H_2S$-limiting treatments, accompanied by a relative decrease in expression of sulfur oxidation genes (Fig. 2b).

### DE of key genes involved in the CBB and rTCA cycles

The gene *cbbM* encodes RuBisCO (form II), the carboxylating enzyme of the CBB and ATP citrate lyase (encoded by *aclAB*), which is one of four key enzymes needed to run the TCA cycle in reverse. *aclA* and *cbbM* were selected to represent pathway changes. When dissolved $O_2$ was limited, *cbbM* showed a decrease in DE. Conversely, *aclA* exhibited relative increases in DE when dissolved $\sum H_2S$ was limited or absent (Fig. 2c). Some comparisons showed similar responses for both genes, such as decreased DE when $NO_3^-$ was limiting but $\sum H_2S$ and $O_2$ were replete, and increases in DE of both genes at $\sum H_2S$-limiting conditions compared with $\sum H_2S$-replete conditions (and in the absence of $NO_3^-$) (Fig. 2c).

### Evidence of energy conservation linked to the rTCA

Genes for ACL and 2-oxoglutarate ferredoxin oxidoreductase (OGOR, encoded by *korABCD*) are essential enzymes of the rTCA cycle. They are found in a gene cluster that also contains a putative flavin-based electron bifurcating (FBEB) complex (Hdr-Flx), which is situated between *aclAB* and *korABCD*, other rTCA/TCA genes and a transhydrogenase (encoded by *pntAB*). This cluster exhibits gene expression patterns that suggest they constitute a functional operon, with similar DE patterns across treatments (except for genes encoding thiol-fumarate reductase (*tfrAB*)). Specifically, these genes 'increase' in expression when dissolved $H_2S$ is limited and 'decrease' in expression when dissolved $NO_3^-$ is limiting (Extended Data Fig. 3a). We also see co-expression of the intergenic regions between the genes along this gene cluster, further supporting that it is an operon (Extended Data Fig. 3b). These data provide empirical evidence for an existing theoretical metabolic model[11], which posits that FBEB using Hdr-Flx may be involved in rTCA carbon fixation where electrons from NADH are shuttled into the rTCA via Hdr-Flx either directly to thiol-fumarate reductase and OGOR or by way of $Fd_{red}$ and a DsrC-like protein.

### Metabolic alliances revealed via network analysis.

Module membership and network visualization revealed a clear grouping of genes with associated metabolic systems. Foremost, genes from the rTCA gene cluster, along with hydrogenases, type 2 V-type ATPases and genes associated with denitrification grouped together in the gold and teal modules. The genes for the CBB cycle, reverse dissimilatory sulfate reductase system (rDSR), sulfur oxidizing system (Sox) and periplasmic sulfide oxidation (such as *fccA* and *sqrA*) grouped together in the pink and cherry modules. Most of the genes for motility were clustered in one module (grey60) and shared only few connections with the rest of the network (Fig. 3). The preprocessing and analysis of data structure for this co-expression network can be found in Extended Data Fig. 4a–f.

### DE in network and hub genes

DE followed network topology (Extended Data Fig. 5a–d); for example, when hub genes had DE, there were concomitant responses in neighbouring genes. The hub genes of the rTCA, FBEB system, PntAB and a group 1e hydrogenase (Hyd1e) all had increased DE under $\sum H_2S$

and $O_2$ limitation (Fig. 2a and Supplementary Table 2). The CBB gene for ribulose-phosphate 3-epimerase (Rpe, encoded by *rpe*) is a hub gene (notably, *cbbM* is not). Rpe catalyses the interconversion of Ru5P and Xu5P, a key step in the regeneration of RuBP. While it can also be involved in the PPP, other genes involved in the CBB (*cbbM*, *tkt*, *gap* and *prkB*) followed the pattern of this hub gene by also showing a decrease in DE at low $O_2$ conditions. Under limited dissolved $\sum H_2S$, key electron transport chain (ETC) genes—*nuoL* (NADH dehydrogenase, complex I of the ETC) and *petB* (cytochrome $bc_1$, complex III)—show opposite expression patterns, with *nuoL* increasing and *petB* decreasing. *petB* is in the pink module, along with CBB and sulfide oxidation genes, while nuoL is part of the light blue module, linking to the gold and teal modules associated with the rTCA and Hyd1e pathways. Genes of the rDSR (involved in sulfide oxidation) were relatively decreased under limiting conditions of oxygen and sulfide (similar to their hub neighbours *petB*, *rpe*, *sqrA* and *fccA*). The genes involved in denitrification (*norCB*, *nosZ* and *nirS*) had higher levels of expression under sulfide limitation (similar to their hub neighbours in the rTCA). For a complete list of genes, their DE patterns and network relationships, see GEO series accession no. GSE249345.

### Analysis finds bi-modal distribution of metabolic processes

In the co-expression network, rTCA genes had 148 first neighbours, while CBB genes had 520, with only one shared between them (a gene of unknown function). Other metabolic systems shared first neighbours with either only the CBB or the rTCA. For example, genes associated with assimilatory nitrate reduction had first neighbours with the CBB, while those involved in dissimilatory nitrate reduction had first neighbours with the rTCA. The transmembrane bound nitrate reductase (Nar) and periplasmic nitrate reductase (Nap) can both be involved in assimilatory or dissimilatory nitrate reduction[23,24]. However, in the network, the narGHI genes shared first neighbours with genes that encode GOGAT (glutamate synthase, which is a key assimilatory enzyme), and the Nap genes shared first neighbours with *nosZ*, *norCB* and *nirS*, which function in the dissimilatory reduction of nitrite to $N_2$. These data support a model whereby the Nar complex is operating in assimilatory nitrate reduction and the Nap complex is being used for dissimilatory nitrate reduction (Figs. 4a and 5). Most sulfur oxidation genes shared first neighbours with the CBB and not the rTCA (genes encoding Sox and rDSR, as well as *aprA*, *aprB*, *sat*, *sqrA* and *fccA*). An exception to this was seen in genes for sulfur globule proteins (Sgp) and genes for the membrane-bound sulfite-oxidizing enzyme (Soe), both first neighbours with the rTCA (Figs. 4b and 5). Both [NiFe] hydrogenases (1e and 3b), V-type ATPases, as well as a putative Mrp−Mbx complex (involved in zero valent sulfur ($S^0$) reduction that is coupled to the production of a sodium motive force in other organisms[25]) shared first neighbours with the rTCA and not the CBB. Conversely, genes for cytochrome $bc_1$ (encoded by *petABC*), a cytochrome $cbb_3$ oxidase (COX) (complexes III and IV of the ETC), and F1-ATPase shared first neighbours with the CBB and not the rTCA (Figs. 4c and 5). The systems with genes that had first neighbours to both carbon fixation pathways are the Qmo (which is thought to transfer electrons from sulfite to the quinone pool)[26], complex I of the ETC and some amino acid synthesis pathways (Figs. 4b and 5). However, when a system shared first neighbours with both, they had different genes (subunits) that are neighbours. In pathways for the biosynthesis of amino acids, the lysine, asparagine, ornithine, shikimate and phenylalanine/tyrosine pathways had genes with neighbours to both the CBB and the rTCA (Figs. 4d and 5). Endoriftia had genes encoding multiple isoforms of PFOR and OGOR, most of which shared first neighbours with either the rTCA or the CBB, which may offer a hint at in vivo function/direction (Fig. 5).

### Module−condition correlations

The correlation between modules and vessel conditions was strongest with oxygen and sulfide (Extended Data Fig. 6). Modules gold and teal

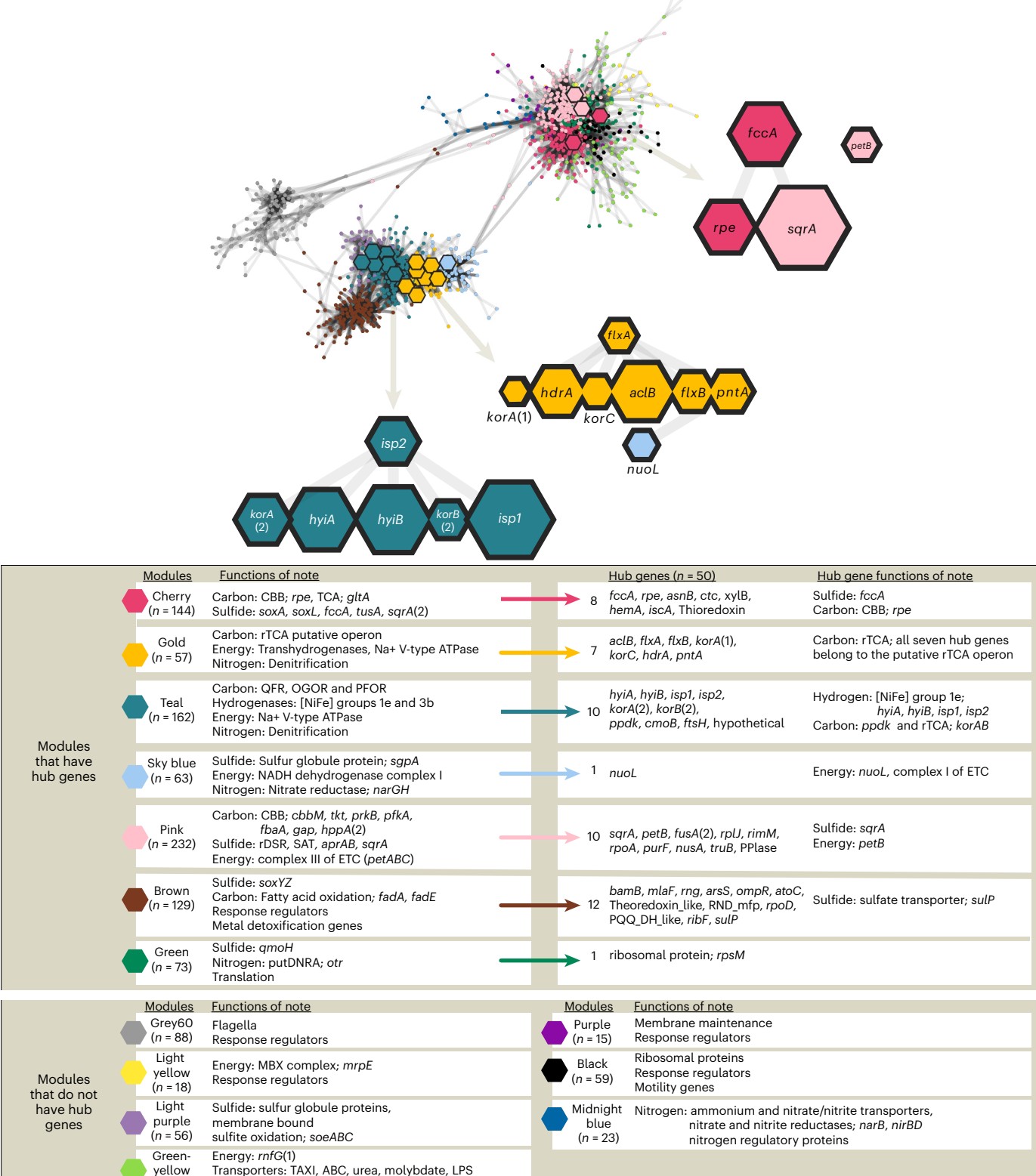

**Fig. 3 | Visualization of network and hub genes made with WGCNA and Cytoscape.** The network (top) was made using these filters: edge weight filter >0.05, degree >2 and a nearest neighbour of 10, leaving 1,194 nodes (genes) and 16,271 edges in the network. Each hexagon represents a node, the connections are edges, with hub genes indicated by larger hexagons. Hub genes that were involved in metabolic pathways of interest were zoomed out, proportional to the MCC value. Colours indicate the module grouping for each node. Metabolic functions of note are listed for each module, along with the corresponding hub genes for that module (if present).

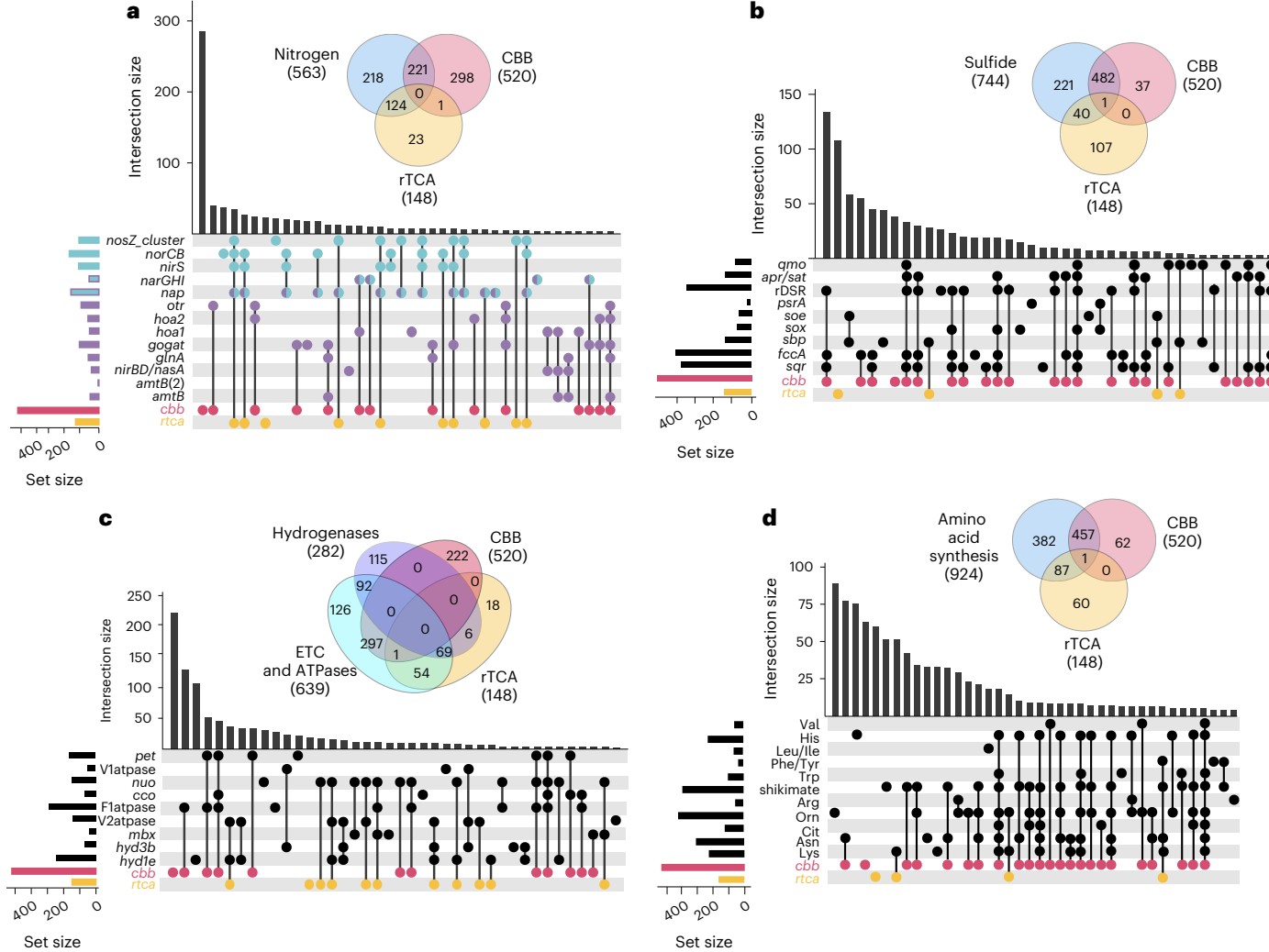

**Fig. 4 | First-neighbour intersections between the CBB, the rTCA and other metabolic functions in the co-expression network.** Venn diagrams illustrate the shared first neighbours between broad functional categories and the rTCA and CBB. Beneath each diagram, intersecting set plots detail contributions from pathways and/or functions. Cherry dots signify the CBB, while gold dots denote the rTCA. Columns with overlapping dots highlight intersections of shared neighbours in the network. **a**, Nitrogen metabolism intersections, where purple dots represent assimilatory processes, blue dots symbolize dissimilatory processes, and dual-coloured dots indicate possible utilization in both functions. **b**, Sulfide metabolism intersections. **c**, Hydrogen metabolism, ETC and ATPases. **d**, Amino acid metabolism intersections.

showed the highest correlation with dissolved $\sum H_2S$ concentrations, and purple and black with dissolved $O_2$ concentrations (Fig. 6a,b). The most significant genes in the gold and teal modules, responding with an increase in DE under dissolved $\sum H_2S$ limitation, were genes encoding (1) rTCA (along with Hdr-Flx and *pntAB* in that gene cluster), (2) Hyd1e, (3) a membrane-bound quinol:fumarate oxidoreductase (encoded by *frdCAB*), which could also act as a succinate dehydrogenase (direction currently unknown), and (4) nitric oxide, nitrate and nitrite reductases (Nor, Nir and Nap). Conversely genes encoding a dimeric OGOR (*korAB*), a possible pyruvate ferredoxin oxidoreductase 'PFOR-like', a cytoplasmic nitrite reductase (*nirB*) and a nitrogen regulatory gene *glnL* all displayed a decrease in expression (Fig. 6c, and Supplementary Tables 4 and 5). The most significant genes within the black and purple modules, responding with an increase in DE, were genes encoding (1) Hyd1e, (2) enzymes involved in pyruvate conversions: a pyruvate phosphate dikinase (*ppdk*), a pyruvate ferredoxin oxidoreductase (PFOR(3), the number added to distinguish from other isoforms in the genome), (3) glycogen catabolism (*malQ*), and (4) a sulfur globule protein (*sgpA*). By contrast, a TCA cycle gene (*acnB*), a gene involved in polysaccharide degradation (GH16), and a nitrogen

storage gene (*cphA*) showed a relative decrease in expression under $O_2$ limitation (Fig. 6d and Supplementary Tables 6 and 7).

## Discussion

The discovery that a chemoautotrophic microorganism's genome harbours the complete pathways for two disparate carbon fixation pathways led to many questions about the utility of having two such pathways[27]. Previous studies established that symbionts express both rTCA and CBB genes and proteins[7,28,29], that individual symbiont cells are likely expressing both pathways, as well as spatial differences in expression within the worm host[30].

We conducted extensive high-pressure incubations of live *Riftia* across a range of environmentally relevant geochemical conditions to assess the metatranscriptomic and metabolic responses of Endoriftia. This study directly measured carbon incorporation into the host and symbiont and revealed net carbon incorporation rates as high as 24 μmol $g^{-1}$ $h^{-1}$, rates that are higher than in most chemolithoautotrophic bacterial communities[31], equal or greater than hydrothermal vent free-living microbial communities[32], and on par with highly productive photosynthetic organisms[33–35].

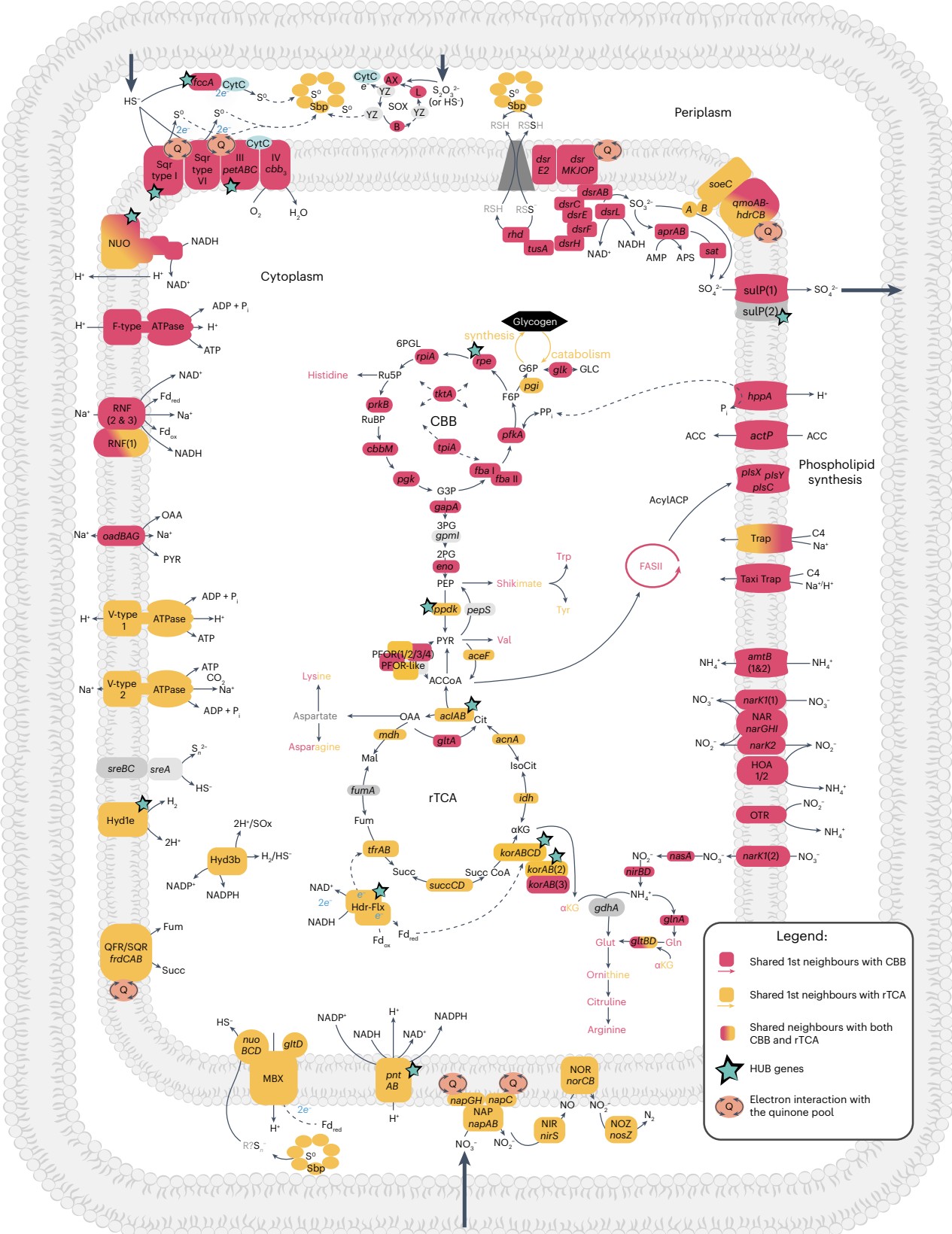

**Fig. 5 | Co-expression patterns in Endoriftia carbon and energy metabolism.**
Gene colours indicate neighbours to the rTCA cycle (yellow) or CBB (red); stars mark hub genes. Genes/complexes of unknown function/direction in vivo: Hdr-Flx, Hyd1e, Hyd3b, MBX, HOA, OTR, QFR/SQR, RNF, OGOR/PFOR isoforms and *sreABC*. Metabolite abbreviations: 6PGL, 6-phosphogluconolactone; Ru5P, ribulose 5-phosphate; RuBP, ribulose-1,5-bisphosphate; G3P, glyceraldehyde 3-phosphate; F6P, fructose 6-phosphate; G6P, glucose 6-phosphate;

GLC, glucose; 3PG, 3-phosphoglycerate; 2PG, 2-phosphoglycerate; PEP, phosphoenolpyruvate; PYR, pyruvate; AcCoA, acetyl coenzyme A; OAA, oxaloacetate; Cit, citrate; Mal, malate; Fum, fumarate; IsoCit, isocitrate; aKg, alpha-ketoglutarate; Suc, succinate; SucCoA, succinyl coenzyme A; Glut, glutamate; Gln, glutamine; Tyr, tyrosine; Trp, tryptophan; Val, valine. Illustration credit: Daria Chrobok, DC SciArt.

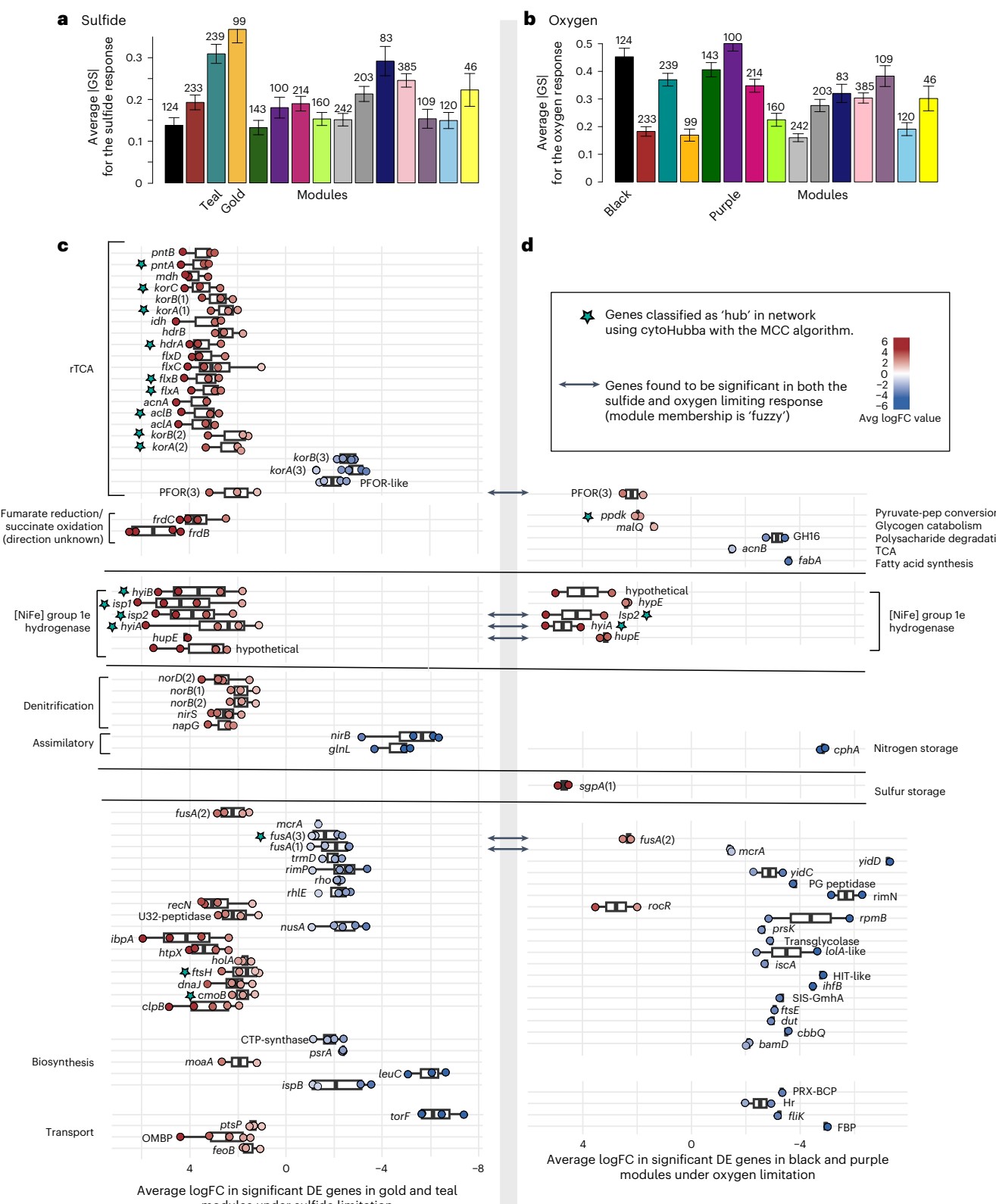

**Fig. 6 | Gene expression patterns for genes most significantly associated with the sulfide or oxygen limitation response. a,b,** Using Pearson's correlation, the average absolute values of gene significance |GS| correlated with sulfide (**a**) and oxygen (**b**) across modules. **c,** DE patterns of the most sulfide-condition-associated genes in the gold and teal modules; red/blue dots indicate relative increase/decrease, with each dot representing a significant DE pairwise comparison. Error bars represent the standard deviation of the mean of all

sulfide-limiting pairwise comparisons that showed significant DE ($P_{adj} \leq 0.05$). **d,** DE patterns for the most oxygen-condition-associated genes within the black and purple modules; red/blue dots indicate relative increase/decrease. Error bars denote the standard deviation of the mean of all oxygen-limiting pairwise comparisons that showed significant DE. See Supplementary Tables 4–7 for mean values, standard deviation and *n* (number of comparisons used in the mean) for each gene found to be significant in these modules.

A co-expression network revealed a metabolic structure with the rTCA and CBB coupled to different and distinct metabolic processes. Notably, the CBB appears to be allied to (broadly speaking) aerobic processes, and rTCA allied to anaerobic processes. Of the 668 genes that share first neighbours with the CBB or the rTCA, only one is neighbours to both, suggesting regulatory independence. Notably, the co-expression patterns of many genes involved in energy transfer/electron flow, sulfide oxidation, and nitrogen metabolism are clearly linked to either the rTCA or the CBB cycle, but not both (Fig. 5).

Co-expression analysis identified genes likely to be the most significant and biologically relevant response for each environmental variable. rTCA genes and Hyd1e appeared to be important responses when carbon fixation was limited by reducing dissolved $\sum H_2S$, with Hyd1e also significant with reduced dissolved $O_2$, suggesting a prominent role when substrate concentrations limit energy availability for endosymbiont metabolism. Of note, sulfide oxidation and CBB genes were not found to be among these genes. Other genes that showed a significant increase in DE during substrate limitation include those encoding *frdCAB*, and PFOR(3). By contrast, there was a concomitant decrease in DE of PFOR-like and *korAB*(3), which may reflect a shift in isoform expression that could have an important role in maintaining metabolic function in different substrate regimes. Another important response to dissolved $\sum H_2S$ limitation appears to be an increase in DE of denitrification genes, along with a decrease in DE of genes involved in assimilatory nitrogen pathways.

While this study sheds light on the relationships between carbon, sulfide, oxygen and nitrogen metabolism, many aspects of the symbionts' physiology are still unclear. Nowhere is this uncertainty more evident than in the role of hydrogenases in this symbiosis. A previous study[22] showed that the provision of dissolved hydrogen does not support net carbon incorporation by the intact symbiosis. However, hydrogenase genes were expressed under all conditions, and all four subunits are 'hub genes' that appear to be closely connected to rTCA genes. These two [NiFe] hydrogenases (Hyd1e and Hyd3b) have been linked to sulfur metabolisms[36–38]. In many organisms, Hyd1e transfers electrons to polysulfide or sulfur reductases (encoded by *psrABC* and *sreABC*, respectively), resulting in ATP generation and $H_2S$ formation (or vice versa)[39–41]. The Hyd3b is a cytosolic NADPH-dependent and bidirectional enzyme with many putative functions: redox pool balance, fermentative $H_2$ evolution via glycogen or pyruvate, and $H_2$ or $H_2S$ evolution via $S^0$ depending on conditions[42,43]. One possibility is that hydrogen is evolved under energy limiting conditions. This phenomenon is seen in cyanobacteria when they break down their glycogen stores during dark fermentation[44]. In addition, it is possible that internal hydrogen cycling is occurring (in which $H_2$ is evolved by one hydrogenase and oxidized by the other), as seen in other organisms[45–47].

The canonical genes for sulfide oxidation with oxygen, and genes involved in generating a proton motive force that drives ATP (*petABC*, and genes encoding CBB3 and F-type ATPase) all cluster with the CBB. This is consistent with the CBB cycle being the primary symbiont carbon fixation pathway; however, $\delta^{13}C$ evidence indicates a mixed carbon fixation strategy, and genes for both pathways are highly expressed (this study and refs. 7,28,29). We also see genes for the rTCA, the FBEB Hdr-Flx complex, a Hyd1e and a transhydrogenase appearing as hub genes in our network analysis, which underscores the importance of these systems in Endoriftia.

The role of oxygen in this symbiosis also warrants further exploration. We know that large quantities of oxygen are consumed by the host and symbiont when dissolved sulfide is abundant (this study and refs. 20,21,48). Yet, we do not know the dissolved oxygen regime inside the trophosome, where microoxic conditions could occur due to the presence of high-affinity haemoglobins and oxygen demand. Because the rTCA relies on oxygen-sensitive $Fd_{red}$, and we see the complete denitrification pathway clustering with the rTCA, this study supports the idea that the rTCA is active in parts of the trophosome lobules with lower $O_2$. These data point to a partitioned metabolism where sulfide oxidation and carbon fixation occur via the CBB cycle, while the rTCA cycle is active and concurrently fixing carbon, with modulations of expression occurring under differing redox regimes. This strategy may enable the host–symbiont association to have high productivity rates and maintain autotrophic poise in the stochastic vent environment.

Finally, modularity in co-expression networks, such as seen in our data, is thought to emerge in organisms that encounter environmental stressors, where this reorganization of the network may increase robustness to perturbation[19]. In fact, some of the most central genes in the network related to substrate limitation were genes of the rTCA cycle and the Hyd1e, underscoring their importance in supporting metabolism during periods of redox stress. Further experimental studies are needed to understand what these roles may be. Nevertheless, we posit that this mode of autotrophy may represent a new carbon fixation modality that confers a fitness advantage in a highly dynamic environment where redox conditions are continuously changing. Continued investigation of these processes could provide deeper insights into the evolution of carbon fixation pathways, as well as the evolutionary histories of microorganisms that harbour multiple pathways[20].

## Methods

### Research expeditions, study sites and *Riftia* collections

The data presented here were generated using *Riftia* collected during two research cruises on board the RV *Atlantis*, the first in November 2014 (AT26-23) and the second in October 2016 (AT37-04). Collections took place during dives with the Human Occupied Vehicle (HOV) *Alvin* at the East Pacific Rise (EPR) vent sites 'Crab Spa', 'Tica' and 'Bio9', all of which are areas of active basalt-hosted hydrothermal vents located near 9° 50′ N, 104° 18′ W at ~2,500 m water depth. These sites are characterized by hydrothermal fluid having elevated $\sum DIC$, $\sum H_2S$, modest concentrations of dissolved $NO_3^-$, very little $NH_4^+$ and variable amounts of $H_2$, surrounded by well-oxygenated bottom water[29,49]. For the experiments, small to moderately sized *Riftia* (≤30 cm in length) were collected towards the end of each dive to minimize their time in non-vent conditions, and brought to the surface via a thermally insulated container on the submersible.

### Replicating the *Riftia* in situ energetic landscape

Upon recovery, *Riftia* that were responsive to touch were quickly placed into high-pressure aquaria within the high-pressure respirometry system (HPRS).

The HPRS includes four acrylic gas equilibration columns that are filled with 0.2-μm-filtered seawater then bubbled with select gases ($H_2$, $H_2S$, $CO_2$, $O_2$, with $N_2$) to re-create vent-like fluids. This fluid feeds the custom-built 2.5 l titanium aquaria via high-pressure pumps (Lewa America) where pressure is maintained at ~20 MPa via back-pressure relief valves (Staval). Flow rates are maintained at ~50 ml min⁻¹. For biological replication, 4–6 tubeworms were placed into each high-pressure aquarium per treatment. Worms were given 12–24 h to acclimate in aquaria before experimental conditions were started. To simulate the observed differences in hydrothermal fluid composition, different treatments were run in which we incubated worms in 0.2-μm-filtered sterilized seawater with various dissolved concentrations of $\sum H_2S$, $O_2$, $NO_3^-$ and $H_2$ (Fig. 1d,e and Supplementary Table 1). These treatments can be generalized into (1) 'replete' conditions, in which $\sum H_2S$, $O_2$ and sometimes $NO_3^-$ were abundant in the aquaria seawater; (2) 'limiting' conditions, in which one or more of these substrates were deficient or nearly absent in the aquaria seawater; and (3) 'controls', in which the animals were incubated without any substrate amendments to the seawater to simulate the cessation of venting (conditions are shown in Supplementary Table 1). Across all these treatments, pH and $\sum DIC$ were held steady at ~6.5 and 4–6 mM, respectively. Worms were maintained in the aquaria at experimental conditions for 2–3 days. It is worth noting that since multiple *Riftia* were incubated in each vessel, the treatments

were not applied independently, introducing a possible spillover effect. Given the difficulty in working with deep-sea organisms, as well as the lifestyle of these worms, we felt that this was the best method because (1) limited time at sea meant it was impractical to incubate 30 worms in separate aquaria; (2) aquaria seawater conditions were effectively steady state, thus minimizing spillover effects due to geochemistry; and (3) *Riftia* is found in very tightly packed clumps, and we believe that putting them together more closely mimics in situ conditions. Finally, this study focuses on the symbiont population within each *Riftia*, wherein each worm already governs the conditions around the symbionts.

To measure the consumption of dissolved gases by the tubeworms during the experimental treatments, fluid was collected before entering the aquaria (incurrent) and upon exiting the aquaria (excurrent). Dissolved $H_2S$ was measured by collecting and preserving incurrent and excurrent fluid subsamples with 2 mM zinc acetate solution for subsequent analyses using a colorimetric sulfide quantification assay (LaMotte, with absorbance read at 670 nm on a Spectramax i3 plate reader). Dissolved $H_2$ was measured with a Unisense $H_2$ minisensor 500 flow cell (range = 0–800 μM; detection limit 0.3 μM) and dissolved $O_2$ was measured with a Presens oxygen FTC-SU-PSt3-S flow cell (range = 0–1,400 μM) by placing both sensors in line with the incurrent and excurrent fluid lines.

During the 2014 cruise, $\sum H_2S$ was consistently measured, whereas during the 2016 cruise, $O_2$ and $H_2$ were also measured as described above. Upon termination of the experiments, the aquaria were depressurized and *Riftia* were weighed on a motion-compensated shipboard balance[50], dissected at 4 °C, and sampled for metatranscriptomic sequencing and isotope analyses (described in the sections below). Uptake rates of $\sum H_2S$, $O_2$ and $H_2$ were calculated as (intake − outtake)/ (total biomass in aquaria × hour)[51]. To compare uptake rates among different treatments, a generalized least square model in R was used that accounted for unequal variances (as seen between high and low amounts of substrates) and allowed for dependence between timepoints that were closer together. The least square (LS) means of these uptake rates were compared using a pairwise-adjusted Holmes test (two-sided), with an α value of 0.05.

### Determining individual tubeworm carbon fixation rates

To robustly determine the rate of inorganic carbon fixation and incorporation into biomass during experimental treatments, a stock solution of 99% $NaH^{13}CO_3$ was added to the intake seawater reservoir to achieve a final isotopic abundance of 2.64% $^{13}C$ of the $\sum DIC$. During the course of the experiments, multiple intake and outtake fluid samples were collected and filtered using 0.2 μm disposable filter capsules, then stored in vacuum-evacuated Exetainers (Labco) for later isotopic analysis where stable isotope ratios in each sample were measured using a Deltaplus XP mass spectrometer at the Yale Analytic and Stable Isotope Center (YASIC).

After cessation of experiments, symbiont-containing tissues (the trophosome) and non-symbiont host tissues (gill and skin) from each worm were frozen at −80 °C for later isotopic analyses (Fig. 1f). In the laboratory, the tissues were lyophilised using a FreeZone 2.5 freeze dryer (Labconco) to minimize carryover of inorganic carbon; all lyophilised samples were bathed in 0.1 N HCl for ~10 min, rinsed in deionized water and then dehydrated in a vacuum oven (Labconco) at 50 °C. Once dried, the tissues were finely ground using a glass mortar and pestle and dispatched to the Boston University Stable Isotope Laboratory. There, samples were placed into tin capsules and precisely weighed using a microelectronic balance, and then combusted in a carbon and nitrogen analyser (Eurovector). The resultant gases were separated by chromatography and analysed using a GV Instruments IsoPrime isotope ratio mass spectrometer. The ratios of $^{13}C$ to $^{12}C$ isotopes were determined against international standards such as NBS 20 (Solenhofen Limestone), NBS 21 (spectrographic graphite)

and NBS 22 (hydrocarbon oil). The $\partial^{13}C$V-PDB values (per mille) were calculated using equation (1), with $R$ being the atomic ratio of $^{13}C/^{12}C$.

$$\partial^{13}C = \left(\frac{R_{\text{sample}}}{R_{\text{standard}}} - 1\right) \times 1,000 \qquad (1)$$

The method used to calculate the incorporation rates of DIC into tissues was adapted from ref. 51. This involved computing the atomic percentages ($A\%$) for both labelled and naturally abundant samples, using their $\delta^{13}C$ values such that:

$$A\% = \left(\frac{R_{\text{sample}}}{R_{\text{sample}} + 1}\right) \times 100 \qquad (2)$$

To calculate the percentage of $^{13}C$ that was incorporated ($\%^{13}C_{\text{inc}}$) into biomass during the course of the experiments, the $A\%$ of the tissue from worms that were not exposed to the isotopic label ($A\%_{\text{nat}}$) was subtracted from the $A\%$ of tissue from experimental worms exposed to labelled inorganic $^{13}C$ ($A\%_{\text{lab}}$), and this was divided by the $A\%$ of $DI^{13}C$ of the fluid ($A\%_{\text{wat}}$) the experimental worms were incubated in (for example, how much isotopic label they were exposed to) subtracted by the $A\%_{\text{nat}}$ (see below):

$$\%^{13}C_{\text{inc}} = \left(\frac{A\%_{\text{lab}} - A\%_{\text{nat}}}{A\%_{\text{wat}} - A\%_{\text{nat}}}\right) \qquad (3)$$

The total weight of $^{13}C$ incorporated ($W^{13}C_{\text{inc}}$) was calculated by multiplying the $\%^{13}C_{\text{inc}}$ by the total dry weight of the tissue analysed, and this value was used to calculate the rate of carbon incorporated per dry weight amount (Dry$C_{\text{inc}}$) expressed as μM $^{13}C$ g$^{-1}$ h$^{-1}$, where MW stands for the molecular weight of $^{13}C$:

$$\text{Dry}C_{\text{inc}} = \frac{[(W^{13}C_{\text{inc}}/MW) \times 1,000]}{(DW \times \text{hours})} \qquad (4)$$

This Dry$C_{\text{inc}}$ was converted into a wet weight $C_{\text{inc}}$ rate by multiplying this value by the dry weight (DW) to wet weight ratio for each sample. The $C_{\text{inc}}$ values reported in this paper were from the symbiont-bearing tissue (the trophosome). However, the plume and the skin were also analysed. For each tissue type, the $A\%_{\text{nat}}$ value was calculated by using the mean values for the natural abundance of these tissue types.

The trophosome tissue is ~24% symbionts by volume[52] and is the vascularized organ that contains the specialized host cells that house the symbionts. The $^{13}C$ incorporation rates herein represent the net carbon fixation attributable to symbionts' sulfide-dependent chemoautotrophic carbon fixation. Although there is evidence that the tubeworm can carboxylate pyruvate to a 4-carbon organic acid (such as succinate or malate)[53,54], those rates are insufficient to support net growth. Moreover, they would not likely be stimulated by the provision of sulfide because only the symbionts can use that as an electron donor. Thus, any carbon incorporation due to host carboxylation reactions is represented by the $^{13}C$ incorporation rates measured in the absence of sulfide (that is, the 'no sulfide' conditions), which also does not exclude the occurrence of symbiont autotrophy utilizing elemental sulfur stores under these conditions.

The rate of net inorganic carbon incorporated into biomass ($C_{\text{inc}}$) was calculated as micromoles per gram wet trophosome weight per hour (μmol g$^{-1}$ h$^{-1}$). Rates were compared in R using a linear mixed effects model that accounts for unequal number of *Riftia* per treatment, with $C_{\text{inc}}$ as the dependent variable, aquaria condition and sample replicate as fixed effects, with each worm being a random effect: lmer($C_{\text{inc}}$ ≈ aquaria conditions + sample_rep + (1| wormID)). This model was used to calculate the LS means of $C_{\text{inc}}$. LS means were compared using a pairwise-adjusted Holmes test (two-sided), with an α value of 0.05.

## Symbiont RNA extraction and sequencing

Messenger RNA was sampled from the trophosome tissue, which contains symbionts, of 30 separate worms. For each condition in the aquarium, three worms were used as biological replicates. To quickly stabilize mRNA, symbiont-containing trophosome tissues were immediately homogenized with a Tissue-Tearor homogenizer (BioSpec) in 1 ml of TRIzol reagent (Thermo Fisher), or placed in 5 ml of RNALater (Thermo Fisher), allowed to incubate for ~8 h at 4 °C and then stored at −80 °C. For both RNALater and TRIzol stored samples, total RNA was extracted using the Direct-zol RNA MiniPrep kit (Zymo Research), following manufacturer instructions. Extracted RNA quality was checked using an Agilent Bioanalyzer 2100. Total RNA was normalized and sent to the Microbial 'Omics Core (MOC) at the Broad Institute (Cambridge, Massachusetts), where DNA and ribosomal RNA removal, library prep and sequencing were completed. Complementary DNA libraries were constructed from 0.5 to 1 µg of RNA using a modified RNAtag-seq protocol[55], whereby an adaptor was added to the 3′ end of the cDNA by template switching using SMARTScribe (Clontech) after reverse transcription[56]. cDNA was sequenced with a 2 ×33- to 75-bp paired-end protocol using the Illumina Novaseq 6000 platform.

## Illumina sequence read preprocessing

The quality of raw, unfiltered sequencing reads was assessed using FastQC (v.0118) (https://www.bioinformatics.babraham.ac.uk/projects/fastqc/). This assessment confirmed that, while no aberrant base quality-by-cycle profiles were detected, there was some evidence for retained adaptor sequence, as well as an abundance of over-represented sequences. The latter often reflect undesirable enrichments for non-target sequences such as rRNAs or other ubiquitous non-coding sequences. To the extent that these reads map to annotated transcripts, these reads impact normalization methods and provide an over-optimistic picture of statistical power for downstream differential expression analyses. Thus, reads were subsequently processed to eliminate these biases. First, adaptor sequences were removed with TrimGalore! (v.0.6.5) (www.bioinformatics.babraham.ac.uk/projects/trim_galore/), setting the minimum retained read length, stringency and error rate to 35, 5 and 0.01, respectively. Trimming low quality bases from reads was not undertaken because (1) base qualities of libraries was high overall and (2) quality trimming of reads has been shown to distort expression estimates[57]. Second, over-represented reads were removed (https://github.com/harvard-informatics/TranscriptomeAssemblyTools). Finally, we filtered out reads originating from rRNAs by mapping read pairs to the SILVA rRNA database[57] (release 138) and removing read pairs for which ≥1 read aligned to the database. Specifically, sequences for SSURef NR99 were transformed to DNA space by replacing uracil (U) bases to thymine (T). Reads were then aligned to the database with Bowtie2 (v.2.5.3)[58] in 'very-sensitive-local' mode. Sequencing the combined host/symbiont trophosome tissue using Illumina Novaseq yielded an average of 31 M paired-end reads per sample. After the removal of rRNA and over-represented reads, there were an average of 19 M remaining reads. The percentage of these reads that mapped to the symbiont genome averaged 2.6 M (14%), which is sufficient to detect differential expression with statistical significance[59]. Of the 3,316 genes in the published genome, 3,140 (~94.6%) were sufficiently abundant (defined as greater than one count per million in three or more samples) to be included in the analyses.

## DE analysis

Transcript abundances were first quantified with RSEM[60] (v.1.3.1) against the recently published, complete reference genome for 'Ca. Endoriftia persephone' (RefSeq accession GCF_023733635.1)[61]. DE between specific pairs of conditions was carried out in a linear modelling framework with the limma-voom (v.3.17) package in R (v.4.0–4.3.3)[62,63]. In a comparison of multiple differential expression

tools, limma-voom was a top performer along with sleuth[64]. Within limma-voom, expression estimates were normalized using the TMM method[65], and only genes that showed expression values greater than one CPM in three or more libraries were used before DE testing. Because an initial principal component analysis (PCA) of expression data indicated that some samples appeared to be outliers and did not cluster with other samples derived from the same condition, we took advantage of a unique feature of limma-voom by estimating 'precision weights' for each sample and incorporating these weights into DE testing. These precision weights account for variance between different observations, so that poor-quality samples will be 'down-weighted' in the analysis. Furthermore, this approach precludes the necessity of discarding these samples, which is particularly an issue with bulk RNA-seq experiments for which biological replication is typically low. DE tests were performed by first fitting a linear model, via a design matrix, to the entire dataset. Comparisons between pairs of conditions of interest were then performed by extracting linear contrasts for these comparisons, followed by empirical Bayes moderation of the standard errors. Differentially expressed genes were determined using a false discovery rate (FDR) cut-off of ≤0.05, which was calculated using the Benjamini–Hochberg method, a standard approach for FDR calculations in DE testing frameworks. We note that gene expression data are not a quantitative representation of carbon fixation rates, and transcription does not always correlate with protein abundances; however, previous studies have found a much tighter correlation when looking at population-level abundances after environmental steady states have been reached (as in this study)[66].

## Gene annotation

To identify the function of genes that have unclear annotations, we deployed a variety of approaches using these programmes: HydDB (https://services.bire.au.dk/hyddb/), DeepTMHMM (https://dtu.biolib.com/DeepTMHMM), HMMR (https://www.ebi.ac.uk/Tools/hmmer/), STRING (https://string-db.org/), Metal Predator (http://metalweb.cerm.unifi.it/tools/metalpredator/), NCBI BLAST, NCBI COBALT and CD search (www.ncbi.nlm.nih.gov/home/analyze/).

## Weighted gene co-expression network analysis

Co-expression patterns were analysed using the Weighted Gene Co-Expression Network Analysis (WGCNA) package (v.1.72) in R[67]. This analysis was unsupervised, meaning no previous filtering according to DE or function was performed. Of the 3,140 genes that were used in the DE analysis above, only the top 2,500 genes that displayed the highest variable expression were used.

Standard methods from the WGCNA pipeline were used to construct a co-expression signed hybrid network, with the normalized and weighted gene expression profiles from limma-voom as input and using refs. 67–69. Analysis of the fit index and mean connectivity revealed that a soft threshold power of $\beta = 8$, for a signed hybrid network, adhered to a scale-free topology model the best, in line with biological assumptions. Using this soft thresholding power of 8, genes were clustered into 14 modules after merging highly connected modules. An analysis of co-expression patterns in individual samples with aquaria condition and cruise year was visualized with the sample dendrogram aquaria condition heat map, which shows no apparent outliers and no discernible batch effect by cruise year. The resulting network was visualized and further analysed in Cytoscape (3.9.1)[70]. Before import into Cytoscape, these genes were filtered according to edge weight >0.05, which left 1,945 genes. In Cytoscape, the network was visualized using the perfuse force directed layout. Network scoring to find hub genes was done using cyto-Hubba (v.0.1), a Cytoscape plugin that performs hub object analysis[71] using the maximal clique centrality (MCC) method, as suggested by the authors. Cytoscape was also used to find the nearest neighbours of genes, which were manually grouped according to putative metabolic function. From these

metabolic groups, the intersection between sets of first neighbours was visualized using the R programme 'UpSetR' (v.1.40)[72].

To identify genes that are biologically relevant to the phenotypic response to sulfide and/or oxygen limitation, we used an approach commonly utilized in human disease research to search for candidate drug targets and/or mechanisms[73–76]. This method relates network structure (modules) to external conditions or traits by calculating (1) gene significance (GS), which is the Pearson correlation between a given gene and the external condition variable (that is, sulfide and/or oxygen), and (2) module membership (MM), which is the correlation between each gene and the module eigengene (ME) representing the first principal component of each module. For each condition variable, we looked at the most significant genes within the two most significant modules (calculated by having the highest average gene significance for that variable). The condition variables were assigned binary values for each condition, such that an assignment of one was given to treatments that were replete with sulfide (or oxygen), and zero to treatments that were limiting in that substrate. Genes were classified as significant for that condition variable if their $|GS| > 0.2$, with a $P \leq 0.05$ and their $|MM| > 0.8$. These genes were further filtered on the basis of the limma results for DE for that condition variable. Since the limma DE analysis was calculated by comparing the DE of one treatment against another, there were five sulfide and two oxygen limitation comparisons used to calculate the average logFC for each condition variable (Supplementary Tables 4–7).

## Statistics and reproducibility

*Riftia* used in these experiments were selected on the basis of size (due to size constraints of the aquaria) and responsiveness. After this non-random sampling, *Riftia* were randomly assigned to treatment aquaria. *Riftia* that died during the course of the experiments were excluded from analyses. No other data were excluded from analyses. No statistical method was used to predetermine sample size in aquaria. The investigators were not blinded to allocation during experiments and outcome assessment.

## Reporting summary

Further information on research design is available in the Nature Portfolio Reporting Summary linked to this article.

## Data availability

Raw sequencing data have been submitted to the NCBI Sequence Read Archive (SRA: SRP323622; project ID: PRJNA736714). Processed data files (read counts, differential expression and co-expression analyses) have been deposited in the NCBI Gene Expression Omnibus (GEO) and are accessible through GEO Series accession number GSE249345. All other data are available in the supplementary material. Source data are provided with this paper.

## Code availability

All code used in these analyses are available in GitHub at https://github.com/harvardinformatics/EndoriftiaTranscriptomics or via Zenodo at https://doi.org/10.5281/zenodo.10894444 (ref. 77).

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

## Acknowledgements

We thank K. Scott of the University of South Florida for expeditionary leadership and intellectual support; the crew of the RV *Atlantis* and HOV *Alvin* for help with sample collections and the high-pressure mobile lab operations; J. Sanders, S. Sylva, R. Beinart, A. Gartman, K. Frank and L. Simonson for help with the respirometry system; C. Breusing for intellectual feedback on the manuscript; E. Zinser for assistance with carbon fixation rate comparisons; and past and present members of the Girguis lab for support and feedback. Statistical support was provided by data science specialist S. Worthington at the Institute for Quantitative Social Science, Harvard University. This work was supported by Gordon and Betty Moore Foundation grant no. 9208 to P.R.G., NSF 1940100 to P.R.G., and NASA Astrobiology Program award no. 80NSSC19K1427.

## Author contributions

J.H.M. and P.R.G. conceptualized the project. The experimental design was developed collaboratively by J.H.M., P.R.G. and J.A.D. Experiments were conducted by J.H.M. and J.A.D., along with the associated molecular and geochemical measurements. J.H.M. and A.H.F. analysed gene expression. J.H.M. also performed the geochemical and isotopic rate analyses, as well as the weighted gene co-expression network analysis (WGCNA). The manuscript was written by J.H.M. with contributions from P.R.G., J.A.D. and A.H.F.

## Competing interests

The authors declare no competing interests.

## Additional information

**Extended data** is available for this paper at https://doi.org/10.1038/s41564-024-01704-y.

**Correspondence and requests for materials** should be addressed to Jessica H. Mitchell or Peter R. Girguis.

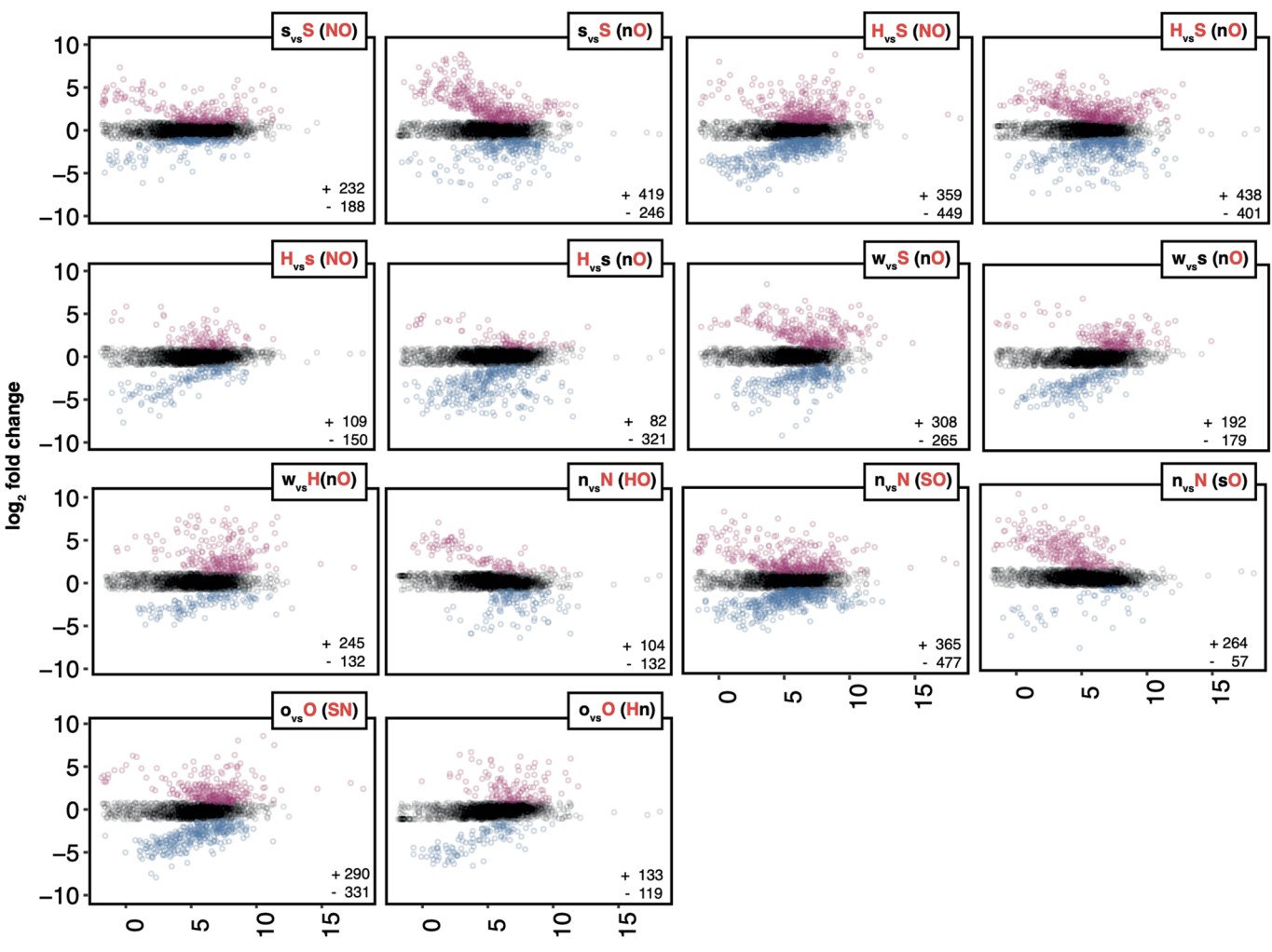

**Extended Data Fig. 1 | MA plots of each pairwise comparison that showed DE represented by log2 fold change vs. counts per million (CPM).** $\log_2$ fold change calculated with limma package in R with a two-sided linear model fit followed by empirical Bayes moderation of the standard errors (using the eBayes function with robust = TRUE parameter), controlling for multiple comparisons with a p-adjusted ≤ 0.05 (using the Benjamini-Hochberg method for false discovery rate,). Dots colored in magenta indicate genes that showed a significant relative increase, and dots colored in blue indicate genes that showed a significant relative decrease in expression between aquaria conditions. Black dots indicate genes that did not show significant DE (each plot represent reads that mapped to 3140 genes). Numbers in the corner of each box represent the total number of genes that were DE with +/− indicating $\log_2$ fold change direction. For details on aquaria conditions, see Fig. 2a and Supplementary Table 1.

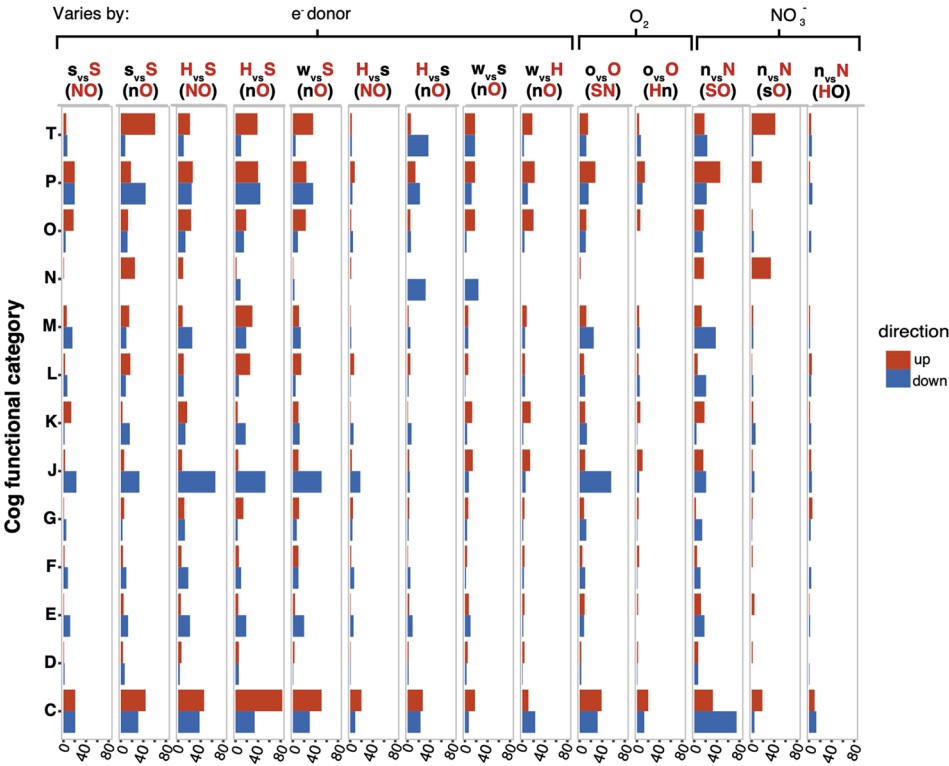

**Extended Data Fig. 2 | Number of genes that showed significant DE in Cog functional categories.** The number of significant DE genes, using a p-adjusted ≤ 0.05, and a log₂ fold change ≥ |−1|, grouped according to Cog functional categories. Red bars indicate the number of genes that showed a relative increase in expression, blue bars indicate the number of genes that showed a relative decrease in expression. Log₂ fold change calculated using limma package in R with a two-sided linear model fit followed by empirical Bayes moderation of the standard errors (using the eBayes function with robust = TRUE parameter) using the Benjamini-Hochberg method for false discovery rate, controlling for multiple comparisons. Comparison of aquaria conditions represented by letters on top row (for abbreviation key see Fig. 1 and Supplementary Table 1). For details on aquaria conditions see Supplementary Table 1. Cog functional category abbreviations listed below: T-Signal transduction L-Replication and repair E-Amino acid metabolism/transport. P-Inorganic ion transport/metabolism K-Transcription D-Cell cycle control and mitosis. O-Post-translational modification/protein turnover J-Translation C-Energy production/conversion. N-Cell motility G-Carbohydrate metabolism/transport. M-Cell membrane biogenesis.

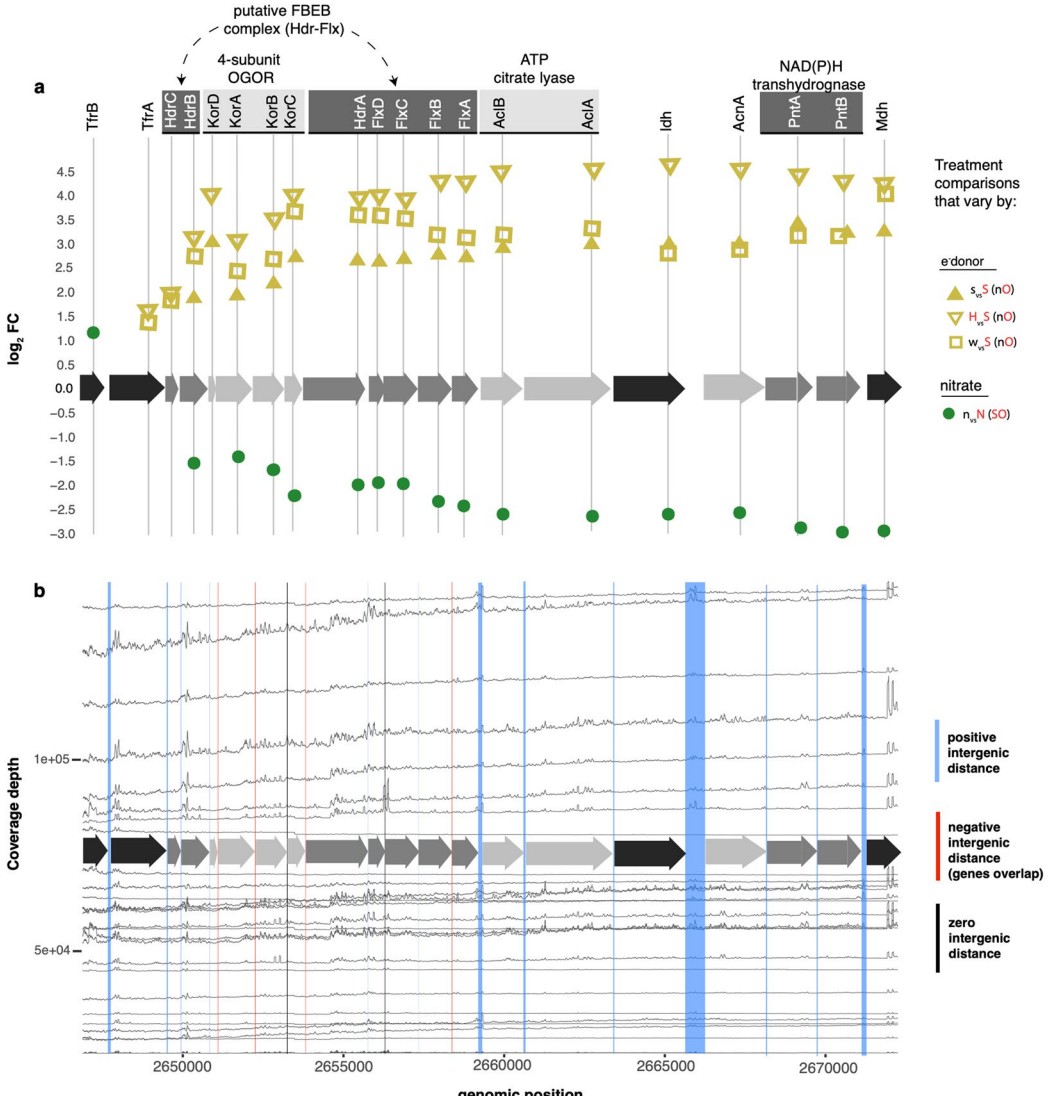

**Extended Data Fig. 3 | Gene location and expression patterns in the putative rTCA operon. a**, The log fold change of genes significantly DE on the conserved gene cluster that contains genes of the rTCA, a putative FBEB system (Hdr-Flx), with a four subunit OGOR (KorABCD) in between HdrA and HdrBC), and a transhydrogenase (PntAB) interspersed with other rTCA genes. Arrows on the y = 0 axis indicate genes in the order they appear in the genome. Colors of the shapes indicate the substrate being varied in the comparison, yellow is limiting or no $\sum H_2S$ compared to $\sum H_2S$ replete and green is no $NO_3^-$ compared to $NO_3^-$ replete. **b**, Coverage depth in all genomic positions within this gene cluster, including intergenic regions. Positive intergenic regions indicated by blue, whereas red lines indicate that the end of one gene overlaps with the start of another, and black lines indicate that there the intergenic distance is zero.

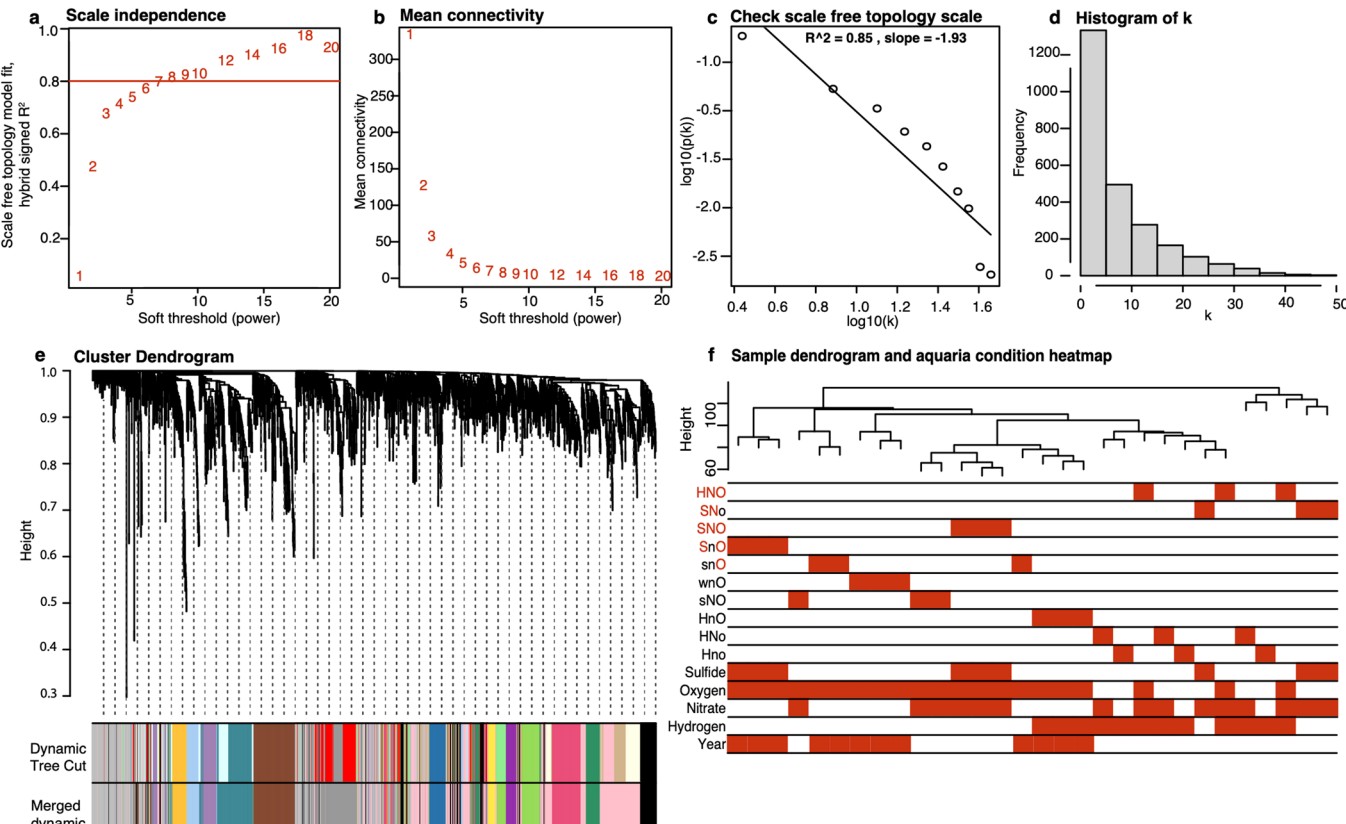

**Extended Data Fig. 4 | Soft thresholding power determination for WGCNA and module and trait clustering of the WGCNA network. a**, The scale free fit index for soft thresholding powers. **b**, Mean connectivity as a function of soft thresholding powers. **c**, Scale free topology of frequency distribution of connectivity p(k) when the soft thresholding power is $\beta = 8$. **d**, Histogram of mean connectivity when the soft thresholding power is $\beta = 8$. **e**, Gene dendrogram with colors along the bottom indicating module color assignment using hierarchical clustering based on TOM dissimilarity (top color row). Bottom color row is module assignment after using a dynamic splicing method, merging highly connected modules with branches reaching below 0.3. **f**, Sample dendrogram and trait heatmap. The leaves of the tree correspond to samples from individual *Riftia*. Red color bands correspond to aquaria conditions in aquaria listed on left (for aquaria conditions and abbreviation key see Fig.1 and Supplementary Table 1). Bottom rows represent conditions merged according to substrate represented by sulfide, hydrogen, oxygen and nitrate. The last row indicates the year these experiments occurred (red represents 2014, white represents 2016).

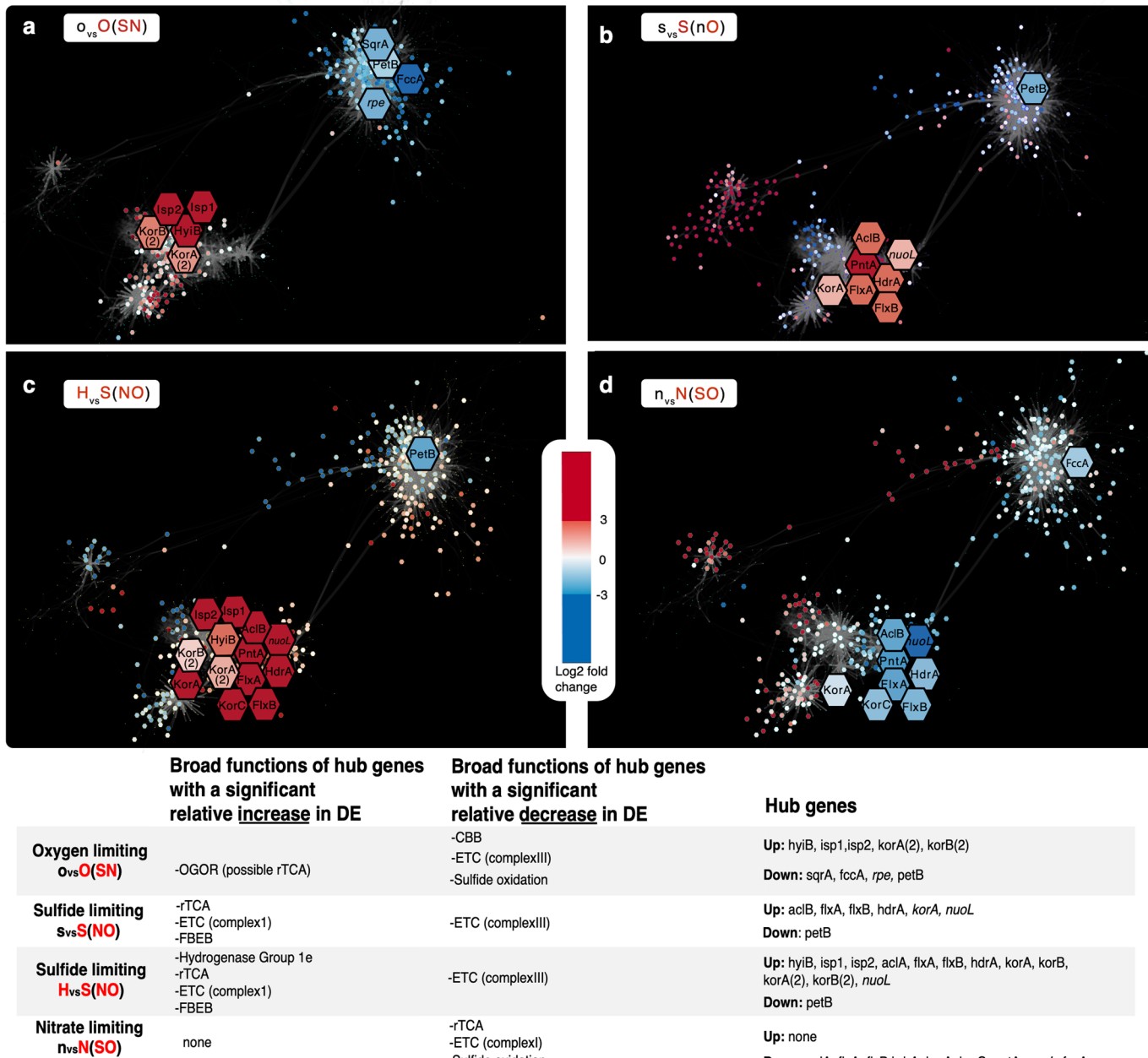

| | Broad functions of hub genes with a significant relative <u>increase</u> in DE | Broad functions of hub genes with a significant relative <u>decrease</u> in DE | Hub genes |
|---|---|---|---|
| **Oxygen limiting** <br> o<sub>vs</sub>**O(SN)** | -OGOR (possible rTCA) | -CBB <br> -ETC (complexIII) <br> -Sulfide oxidation | **Up:** hyiB, isp1,isp2, korA(2), korB(2) <br><br> **Down:** sqrA, fccA, *rpe*, petB |
| **Sulfide limiting** <br> s<sub>vs</sub>**S(NO)** | -rTCA <br> -ETC (complex1) <br> -FBEB | -ETC (complexIII) | **Up:** aclB, flxA, flxB, hdrA, *korA, nuoL* <br><br> **Down:** petB |
| **Sulfide limiting** <br> H<sub>vs</sub>**S(NO)** | -Hydrogenase Group 1e <br> -rTCA <br> -ETC (complex1) <br> -FBEB | -ETC (complexIII) | **Up:** hyiB, isp1, isp2, aclA, flxA, flxB, hdrA, korA, korB, korA(2), korB(2), *nuoL* <br><br> **Down:** petB |
| **Nitrate limiting** <br> n<sub>vs</sub>**N(SO)** | none | -rTCA <br> -ETC (complexI) <br> -Sulfide oxidation | **Up:** none <br><br> **Down:** aclA, flxA, flxB,hdrA, korA, korC, pntA, *nuoL*, fccA |

**Extended Data Fig. 5 | Differential expression patterns overlaid on network in four pairwise comparisons.** Differentially expressed (DE) nodes (genes) in co-expression network in four treatment comparisons where the limiting condition was compared to the replete: **a**, Oxygen limiting **b**, Sulfide limiting **c**, Sulfide limiting with hydrogen and **d**, Nitrate limiting. Only nodes that represented genes DE with a p-adjusted < 0.05 are shown, calculated using limma package in R with a two-sided linear model fit followed by empirical Bayes moderation of the standard errors (using the eBayes function with robust = TRUE parameter) and the Benjamini-Hochberg method for false discovery rate, controlling for multiple comparisons. Red indicates a relative increase in expression and blue indicates a relative decrease in expression using a log2 fold scale. Enlarged nodes indicate relevant metabolic genes that were DE and classified as hub genes in cytoHubba.

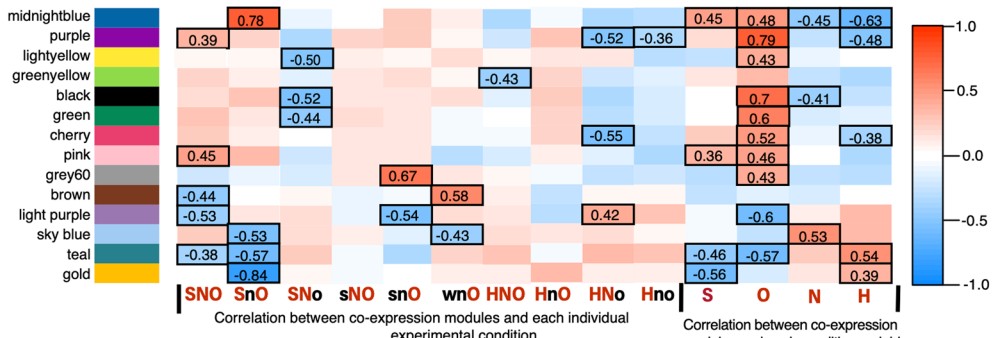

**Extended Data Fig. 6 | A module-condition heatmap showing the Pearson correlation between modules and aquaria conditions.** A Pearson's correlation coefficient was used to find the covariance between modules and aquaria (two-tailed) and p-values calculated using a student asymptotic test. Only correlation values that had a p-value of < 0.05 are shown. Correlation values on the left are for each aquaria condition. Correlation values on the right indicates a specific substrate (S⁻ sulfide, O-oxygen, N-nitrate, H-hydrogen). For example, teal and gold modules are negatively correlated with sulfide replete conditions (S) indicating a pattern of an overall decrease in gene expression of these module's eigengenes.

# Reporting Summary

## Statistics

For all statistical analyses, confirm that the following items are present in the figure legend, table legend, main text, or Methods section.

| n/a | Confirmed | |
|---|---|---|
| ☐ | ☒ | The exact sample size (*n*) for each experimental group/condition, given as a discrete number and unit of measurement |
| ☐ | ☒ | A statement on whether measurements were taken from distinct samples or whether the same sample was measured repeatedly |
| ☐ | ☒ | The statistical test(s) used AND whether they are one- or two-sided<br>*Only common tests should be described solely by name; describe more complex techniques in the Methods section.* |
| ☐ | ☒ | A description of all covariates tested |
| ☐ | ☒ | A description of any assumptions or corrections, such as tests of normality and adjustment for multiple comparisons |
| ☐ | ☒ | A full description of the statistical parameters including central tendency (e.g. means) or other basic estimates (e.g. regression coefficient) AND variation (e.g. standard deviation) or associated estimates of uncertainty (e.g. confidence intervals) |
| ☐ | ☒ | For null hypothesis testing, the test statistic (e.g. *F*, *t*, *r*) with confidence intervals, effect sizes, degrees of freedom and *P* value noted<br>*Give P values as exact values whenever suitable.* |
| ☒ | ☐ | For Bayesian analysis, information on the choice of priors and Markov chain Monte Carlo settings |
| ☐ | ☒ | For hierarchical and complex designs, identification of the appropriate level for tests and full reporting of outcomes |
| ☐ | ☒ | Estimates of effect sizes (e.g. Cohen's *d*, Pearson's *r*), indicating how they were calculated |

*Our web collection on statistics for biologists contains articles on many of the points above.*

## Software and code

Policy information about availability of computer code

| Data collection | No software was used to data collection. |
|---|---|

| Data analysis | FastQC v.0118 (www.bioinformatics.babraham.ac.uk/projects/fastqc/)<br>TrimGalore! v0.6.5 (www.bioinformatics.babraham.ac.uk/projects/<br>trim_galore/)<br> R v4.0.0-v4.3.3<br>Bowtie2 2.5.3<br>ggplot2 v3.2.0 -3.5.0<br>https://github.com/harvardinformatics/TranscriptomeAssemblyTools<br>RSEM v.1.3.1<br>limma v3.17<br>WGCNA v1.72<br>Cytoscape v. 3.9.1<br>cyto-Hubba v. 0.1<br>UpsetR v1.4.0<br>HydDB (https://services.birc.au.dk/hyddb/)<br>DeepTMHMM (https://dtu.biolib.com/DeepTMHMM)<br>HMMR (https://www.ebi.ac.uk/Tools/hmmer/)<br>STRING (https://string-db.org/)<br>MetalPredator (http://metalweb.cerm.unifi.it/tools/metalpredator/)<br>NCBI BLAST<br>NCBI COBALT<br>CD search (www.ncbi.nlm.nih.gov/home/analyze/) |
|---|---|

For manuscripts utilizing custom algorithms or software that are central to the research but not yet described in published literature, software must be made available to editors and reviewers. We strongly encourage code deposition in a community repository (e.g. GitHub). See the Nature Portfolio guidelines for submitting code & software for further information.

## Data

- Accession codes, unique identifiers, or web links for publicly available datasets
- A description of any restrictions on data availability
- For clinical datasets or third party data, please ensure that the statement adheres to our policy

Policy information about availability of data

All manuscripts must include a data availability statement. This statement should provide the following information, where applicable:

Raw sequencing data have been submitted to the NCBI Sequence Read Archive (SRA). SRA: SRP323622. Project ID: PRJNA736714; https://www.ncbi.nlm.nih.gov/bioproject/PRJNA736714. Processed data files (read counts, differential expression, and co-expression analyses) have been deposited in NCBI's Gene Expression Omnibus and are accessible through GEO Series accession number GSE249345; https://www.ncbi.nlm.nih.gov/geo/query/acc.cgi?acc=GSE249345. All other data are available in the supplementary material, and source data files.

## Research involving human participants, their data, or biological material

Policy information about studies with human participants or human data. See also policy information about sex, gender (identity/presentation), and sexual orientation and race, ethnicity and racism.

| Reporting on sex and gender | N/A |
|---|---|
| Reporting on race, ethnicity, or other socially relevant groupings | N/A |
| Population characteristics | N/A |
| Recruitment | N/A |
| Ethics oversight | N/A |

Note that full information on the approval of the study protocol must also be provided in the manuscript.

# Field-specific reporting

Please select the one below that is the best fit for your research. If you are not sure, read the appropriate sections before making your selection.

☐ Life sciences    ☐ Behavioural & social sciences    ☒ Ecological, evolutionary & environmental sciences

For a reference copy of the document with all sections, see nature.com/documents/nr-reporting-summary-flat.pdf

# Ecological, evolutionary & environmental sciences study design

All studies must disclose on these points even when the disclosure is negative.

| Study description | Deep sea hydrothermal vent tubeworms were collected and kept alive at experimental conditions for studying the effect of the external environmental factors (e.g. dissolved gases) on the metabolism of their chemoautotrophic symbionts. |
|---|---|

| Research sample | 30 deep sea hydrothermal tubeworms (Riftia pachyptila) were collected and incubated at in situ pressures and temperatures. During the course of the incubations, aquaria water samples were analyzed for changes in dissolved oxygen, hydrogen, sulfide, dissolved inorganic carbon and pH. The tubeworms were dissected and samples of their trophosome (the symbiont bearing tissue found deep within the worm) were taken for RNA and isotope analyses. Other tissue samples of the worms were also taken for isotope analyses. |
|---|---|
| Sampling strategy | Riftia pachyptila were collected using a human operated vehicle (HOV Alvin) aboard R/V Atlantis. We chose worms ~1ft in length or less so that they would fit in our pressurized aquaria. Worms were collected at the end of the dive with the HOV and upon surfacing, were brought back to pressure within 1-2 hours. Each aquaria held 4-5 worms, in flow through conditions, with active stirring. |
| Data collection | Riftia pachyptila were collected with the submersible Alvin (operated by the pilot) and sampling directions were given by Jessica Mitchell, and site information was recorded for each collection. |
| Timing and spatial scale | Riftia pachyptila were collected during two research cruises to the East Pacific Rise on 02-21 Novemeber 2014 and 08-28 October 2016. Riftia were collected from the vent sites, "Crab Spa", "Tica" and "Bio9" during multiple collections. |
| Data exclusions | Worms that did not present as healthy after incubations were removed from these analyses. Our criteria for healthy are that they must be responsive to touch, responsive to more acute stimuli (e.g., being poked with a toothpick), must not show any body lesions, and must not have an off-odor consistent with tissue necrosis. |
| Reproducibility | All computational analyses were conducted using open source softwares with versions and any flags specified and can be reproduced accordingly. |
| Randomization | A stratified randomization technique was employed when selecting Riftia for incubations such that, each vessel got a similar size distribution of worms. |
| Blinding | Blinding is not relevant for this study because the aim of the study is to characterize and compare the shifts in microbial metabolism under controlled environmental conditions. |

Did the study involve field work?  ☒ Yes  ☐ No

## Field work, collection and transport

| Field conditions | Field work was conducted aboard R/V Atlantis in the East Pacific Rise. Weather reports for each day are available at the RCR repository. Briefly, all samples were collected on clear days with no documented precipitation. |
|---|---|
| Location | Riftia pachyptila were collected during two research expeditions on the R/V Atlantis to the East Pacific Rise using the HOV Alvin in 2014 and 2016 at the following coordinates: 27.4092163l°N, 111.38910334°W. water depth was 1810 meters. |
| Access & import/export | All samples were collected under the appropriate permits, which principally includes US Fish and Wildlife and USDA permits. No animals fall under CITES or the Nagoya protocol |
| Disturbance | Our collection methods are consistent with best practices in the field, and result in minimal or no damage to the vent site when sampling and only animals that were needed for the study were collected. |

# Reporting for specific materials, systems and methods

We require information from authors about some types of materials, experimental systems and methods used in many studies. Here, indicate whether each material, system or method listed is relevant to your study. If you are not sure if a list item applies to your research, read the appropriate section before selecting a response.

## Materials & experimental systems

| n/a | Involved in the study |
|---|---|
| ☒ | ☐ Antibodies |
| ☒ | ☐ Eukaryotic cell lines |
| ☒ | ☐ Palaeontology and archaeology |
| ☒ | ☐ Animals and other organisms |
| ☒ | ☐ Clinical data |
| ☒ | ☐ Dual use research of concern |
| ☒ | ☐ Plants |

## Methods

| n/a | Involved in the study |
|---|---|
| ☒ | ☐ ChIP-seq |
| ☒ | ☐ Flow cytometry |
| ☒ | ☐ MRI-based neuroimaging |

