## [Peer Review File · Nature Microbiology]

Peer Review Information

Journal: Nature Microbiology

Manuscript Title: Co-expression analysis reveals distinct alliances around two carbon fixation pathways in hydrothermal vent symbionts.

Corresponding author name(s): Professor Peter Girguis

Reviewer Comments & Decisions:

Decision Letter, initial version:

Message: 8th August 2023

Dear Pete,

Kyle here, from Nature Micro. I've taken over as handling editor of your manuscript from my colleague Dr. Emily White. Thank you for your patience while your manuscript "Beyond the single pathway: co-expression analysis reveals distinct roles of dual carbon fixation in *Riftia pachyptila* symbionts" was under peer-review at Nature Microbiology. It has now been seen by 3 referees, whose expertise and comments you will find at the end of this email. Although they find your work of some potential interest, they have raised a number of concerns that will need to be addressed before we can consider publication of the work in Nature Microbiology.

The referees do note that your experimental set up is compelling at that the dataset is impressive, however they do note that the conclusions should be better put in the context of the published literature on this topic. The referees, and in particular Reviewer #1, provides excellent direction for some additional analyses, e.g. identifying genes that correlate with the carbon fixation rates, to provide further insight. Other issues raised by the referees that will need to be addressed with revisions include the need for more methods details, how this metabolic strategy benefits symbiosis, and Reviewer #2 also has an additional question about potential pseudoreplication.

Should further experimental data allow you to address these criticisms, we would be happy to look at a revised manuscript.

We strongly support public availability of data. Please place the data used in your paper into a public data repository, if one exists, or alternatively, present the data as Source Data or Supplementary Information. If data can only be shared on request, please explain why in your Data Availability Statement, and also in the correspondence with your editor. For some data types, deposition in a public repository is mandatory - more information on our data deposition policies and available repositories can be found at <https://www.nature.com/nature-research/editorial-policies/reporting->

2standards#availability-of-data.

Please include a data availability statement as a separate section after Methods but before references, under the heading "Data Availability". This section should inform readers about the availability of the data used to support the conclusions of your study. This information includes accession codes to public repositories (data banks for protein, DNA or RNA sequences, microarray, proteomics data etc...), references to source data published alongside the paper, unique identifiers such as URLs to data repository entries, or data set DOIs, and any other statement about data availability. At a minimum, you should include the following statement: "The data that support the findings of this study are available from the corresponding author upon request", mentioning any restrictions on availability. If DOIs are provided, we also strongly encourage including these in the Reference list (authors, title, publisher (repository name), identifier, year). For more guidance on how to write this section please see: <http://www.nature.com/authors/policies/data/data-availability-statements-data-citations.pdf>

* If you have not done so already we suggest that you begin to revise your manuscript so that it conforms to our Article format instructions at <http://www.nature.com/nmicrobiol/info/final-submission>. Refer also to any guidelines provided in this letter.

When submitting the revised version of your manuscript, please pay close attention to our [href="https://www.nature.com/nature-portfolio/editorial-policies/image-integrity">Digital Image Integrity Guidelines](https://www.nature.com/nature-portfolio/editorial-policies/image-integrity). and to the following points below:

2Please use the link below to submit a revised paper:

Note: This url links to your confidential homepage and associated information about manuscripts you may have submitted or be reviewing for us. If you wish to forward this e-mail to co-authors, please delete this link to your homepage first.

Nature Microbiology is committed to improving transparency in authorship. As part of our efforts in this direction, we are now requesting that all authors identified as 'corresponding author' on published papers create and link their Open Researcher and Contributor Identifier (ORCID) with their account on the Manuscript Tracking System (MTS), prior to acceptance. This applies to primary research papers only. ORCID helps the scientific community achieve unambiguous attribution of all scholarly contributions. You can create and link your ORCID from the home page of the MTS by clicking on 'Modify my Springer Nature account'. For more information please visit please visit www.springernature.com/orcid.

If you wish to submit a suitably revised manuscript we would hope to receive it within 6 months. If you cannot send it within this time, please let us know. We will be happy to consider your revision, even if a similar study has been accepted for publication at Nature Microbiology or published elsewhere (up to a maximum of 6 months).

Yours sincerely,

Reviewer Expertise:

Referee #1: marine micro, metabolism, 'omics, SIP
Referee #2: deep sea micro, symbiosis, metabolism
Referee #3: symbiosis, 'omics, metabolism

Reviewer Comments:

Reviewer #1 (Remarks to the Author):

The manuscript of Mitchell et al is an interesting study where Riftia tube worms and their microbial symbionts were subjected to high pressure experiments under different geochemical conditions to test the activity of two different carbon fixation pathways that are encoded by the symbionts genome. They also measured bulk DIC assimilation rates using ¹³C bicarbonate into the host tissue (but it is unclear whether this is the microbial C-fixation rate, see below). As a general comment: there were no line numbers making it difficult to provide comments for specific portions of text and so I tried to be as specific as

3possible but without the line numbers my comments are relegated to being a bit more general in nature.

Its a really impressive experimental dataset with a high amount of replication and controls, and could meet the high mark for publication in Nature Microbiology I think. But there are many points that require clarification before publication could be considered (whether or not DNase treatment was applied, how long the worms were in the aquarium before the experiments were done, see below). Hopefully, the authors can address these points.

I think that the authors actually are under utilizing their dataset, and should do some additional analyses to directly correlate gene expression data with carbon fixation rates under different conditions. This connection was not made as strong as it could be in the manuscript. For example, which genes correlate with higher/lower C-fixation rates? Do some genes increase expression under particular conditions that coincide with higher C-fixation? Were C-fixation rates lower when rTCA expression dominated, and was C-fixation higher when CBB expression dominated? This would go a long way to show which C-fixation pathway was more important and under which conditions. As far as I could tell, the authors primarily correlated the genes with one another (which is really interesting), but they don't correlate genes/pathways with C-fixation rates under different conditions. I think if they can do this it would help alot to show which pathway is more important than the other under different conditions.

Below I have some more specific comments:

Abstract: Is it true that "most" organisms contain only one pathway? There are alot of microbes out there and many of them have two pathways. Is there a citation to support this statement of fact? I think that the authors are probably correct, but still would be nice to have a citation to support this strong statement.

In the second to last sentence of the abstract (and throughout the text) , it is unclear what you mean "allied to".

Introduction first paragraph: I think there are now six carbon fixation pathways known including the novel roTCA cycle (Steffens et al 2021 Nature).

Introduction last paragraph: it is unclear what you mean by "load balancing".

Methods: The worms were collected from a deep sea vent nearly 8 years ago, and it is stated they were placed into a high pressure aquarium afterwards. The authors need to clearly state how long the worms were in the aquarium before the experiments in the manuscript were conducted. Were the experiments in the manuscript done soon after collection or were the worms in the aquarium for several years prior to the experiments? It looks like the experiments were done on the ship and therefore soon after collection, but this was not entirely clear to me and should be stated in the methods.

Please provide full details on how the carbon fixation measurements were done (currently there is only a reference to Mitchell et al 2019 but this is insufficient, all key details should be provided here). How do you differentiate between the carbon fixed by the microbes, and

the carbon that is transferred to the host tissue? Did you sample the microbes and the host tissue separately and make separate measurements? Currently it looks like you only sampled the host tissue, but the host does not fix carbon and so this is not a measurement of carbon fixation its a measurement of how much of the fixed carbon from the symbiont was transferred to the host. Please clarify what exactly was measured (microbial biomass, or host biomass).

How was DNA removed from the RNA extracts prior to sequencing? There is no DNase step reported in the methods, is it true that no DNase treatment was done ? In that case, the transcriptomes are surely contaminated with DNA and cannot be published.

Results:

Please report in the text what the $\delta^{13}\text{C}$ of the Riftia biomass was after the incubations, currently it is only written that it was "far heavier" than the natural abundance. How much heavier? Or is that the "stable isotope ratios" of 40 and 181? Are those $\delta^{13}\text{C}$ values?

It is interesting that you see over expression of cbbM with lower O₂, and higher expression at high O₂. This shows that the prevailing narrative that the CBB cycle is more active at high O₂ is not always true. I think this interesting finding could be highlighted more in the discussion (but it contradicts some statements in the conclusions, see below). Can you conclude why the CBB cycle is responding differently to low O₂ here than in other organisms?

How does the rDSR gene expression change with sulfide concentrations? And, how do the dissimilatory nitrate reductases change with nitrate concentrations? Do they should similar patterns to either the CBB or rTCA expression? It is written that they "share neighbors" with the rTCA. But this does not inform the reader whether they go up or down together or if they are decoupled, and it does not inform whether this is different between treatments.

In the absence of ¹⁵N isotope incorporation measurements, it is unclear to me how the authors can conclude that narGHI is being used for assimilation of nitrate (and not dissimilatory nitrate reduction).

Discussion:

How do the rates of C fixation correlate with the expression of rTCA or CBB under the different treatments? Apologies if I missed it, but I couldn't find any discussion of this.

The statement in the conclusions that "Surprisingly ... CBB (broadly speaking) allied to more aerobic processes and rTCA allied to anaerobic processes." , is a bit of a surprise after reading the discussion and results. That conclusion did not come out clearly in the main text, and seems a bit contradictory with some results (higher cbbM expression at low O₂, higher expression of rTCA at low sulfide, see above). Can you please clarify this point?

"this study has provided the first in depth look at an organism that simultaneously uses two pathways whose substrates and endproducts are allied to different metabolic processes,"

Are you sure you are the first? Giovanelli et al (eLife) looked at co-occurring carbon fixation pathways (rTCA and WLP) in *Thermovibrio ammonienseis*. Moreover, some methanogens have co-occurring WLP and rTCA cycles for carbon fixation.

Reviewer #2 (Remarks to the Author):

This paper describes RNA-level co-expression patterns in *Endoriftia* symbionts of *Riftia* tubeworms, based on the impressive pressure incubations onboard, immediately following animal retrieval from the deep-sea hydrothermal vents. The authors focus on a key question regarding the ability of these organisms to fix copious amounts of carbon, based on the potential use of both the CBB and rTCA cycles by the symbionts, asking how environmental conditions affect these processes. Using coexpression network analyses, the authors identified key modules and hub genes in *Endoriftia* metabolism.

Despite the impressive experimental setup, the findings are not highly novel, as they are often confirmatory of previous work testing the expression of *Endoriftia* proteins in S-rich and S-depleted specimens, and showing the differential expression of CBB and rTCA enzymes (Markert 2007, Leonard et al 2021, Hinzke et al 2021). While this work tests a larger array of conditions, the conclusions mainly highlight the difference between the high and low sulfide conditions. The effects of other conditions were not discussed in detail – for example, Figure 4 is mentioned only briefly.

Another fascinating aspect of this study is the measurements of carbon uptake rates, as well as those of the use of other resources. Yet, the high uptake rates from such experiments were already reported previously by Mitchell et al. 2019, thus the novelty here is also limited.

There is a possible issue of pseudoreplication, given that the experiments were conducted on individuals placed in a single container (is it indeed the case?). This may lead to unknown bottle effects. The authors should at least address this issue, and explain why the results are valid.

Additional topics that require attention include the effects of radial variability in expression patterns (Hinzke et al., 2021), as well as the potential adaptability at the sub-species level (Polzin et al., 2019).

Other comments:

Introduction:

Please fix Hdr-flx to "NADH dehydrogenase/heterodisulfide reductase (Hdr-Flx)"

6Expand the PPP acronym.

Methods:

Can you comment on the potential bottle effects? If I understand correctly, only a single tank was assigned per condition. Wouldn't place individuals in the same tank account as pseudoreplication? For example, see here - [10.1371/journal.pbio.2005282](https://doi.org/10.1371/journal.pbio.2005282).

Oxygen levels in 2014 experiments appear to be very high – how do they compare to the in-situ values? Can you comment on the potential effects of these high values?

Animal sectioning – How the trophosome subsections for RNA extractions were chosen? Could you comment on how your work can cope with the radial variability – e.g., <https://doi.org/10.7554/eLife.58371>. Could this explain the outliers you found in the treatments?

Removal of the overrepresented reads - Wouldn't this affect the data? For example, some genes in tubeworm symbionts, such as those that encode porins and sulfur globules, could have exceptional expression values.

Fix reference placement: database (release 138, 31)

What was the purpose of filtering out the rRNA sequences, given that they don't play a role in the comparative dataset?

Results:

In "Gross patterns in symbiont gene expression" Please rephrase "sufficiently abundant".

In "Evidence of energy conservation linked to the rTCA", please provide evidence for the assumption that the rTCA genes form a functional operon.

Please check for continuous read coverage along the whole operon transcript.

In "First neighbor analysis reveals a bi-modal metabolism" – "A first neighbor analysis was performed on the major metabolic pathways involved in energy metabolism" – energy or carbon?

"Genes related to sulfide metabolism show a clear pattern whereby most sulfide oxidation systems (sox, rDSR, apr/sat, sqrA, and fccA)..." there is a mixture between genes, proteins and systems...

Discussion:

"Surprisingly, these analyses revealed that the rTCA and CBB carbon fixation pathways form metabolic nodes that are coupled to other distinct metabolic processes, which may explain their ability to support such high growth rates in Riftia" – While not done in the same systematic manner as the network analysis employed here, previous studies appear to show some of these, and also potentially contradicting co-expression patterns (e.g., cell size-dependent expression patterns seem to indicate co-expression of RubisCO and

denitrification in Hinzke et al 2021).

Figure 1:

I think it is worth moving column 5 to column 3, to keep similar treatments together.

Please fix (_ indicates seawater only).

In the legend, please expand DE to differential expression.

Figure 2b: Panels B, D – there are additional clusters of upregulated genes – which genes are there?

In the legend, please expand DE to differential expression. ETC?

References styling is inconsistent.

Reviewer #3 (Remarks to the Author):

In their study, Mitchell et al. have investigated the two carbon fixation pathways in the hydrothermal vent tubeworm symbiont *Candidatus Endoriftia persephone*. Using incubation experiments in high pressure aquaria, carbon isotope labelling and transcriptomic analyses, the authors aimed to determine a) how different environmental conditions influence the expression of the two pathways – the Calvin-Benson-Bassam (CBB) cycle and the reductive TCA (rTCA) cycle – and b) whether these two CO₂ fixation pathways have distinct cellular functions, e.g., in providing metabolic precursors – and thus basically c) why the symbiont uses two CO₂ fixation pathways simultaneously.

The study provides some interesting new findings, namely that rTCA and CBB seem to be 'allied' to distinct sets of metabolic pathways, with hardly any overlap between these two hubs, as revealed by co-expression network analyses/first neighbor analysis. This method and its results are indeed innovative and enhance our current understanding of this model symbiont's physiology. Other approaches and findings, such as the observation that rTCA enzymes are particularly abundant when ambient sulfide concentrations are low and energy is limited, confirm previous studies (see below) and are therefore valuable, but not particularly original. The results are of immediate interest to researchers in the field of hydrothermal vent microbiology and physiology, as they add another interesting piece to the puzzle of how gutless tubeworms and their sulfur-oxidizing bacterial symbionts have optimized their extraordinary association. To be interesting to a broader audience, the study would need to elaborate more on how the symbiosis benefits, specifically, from the observed metabolic nodes, one centered around rTCA, the other centered around CBB, (beyond the relatively general assumption of an unspecified 'fitness advantage'). What shapes these 'alliances' between pathways? As it is, the manuscript provides a solid basis for in depth follow-up research – but does, regrettably, not go into further detail to explore and explain individual connections within the nodes.

The manuscript is well written (except for a few minor sloppy mistakes) and clearly comprehensible. It would have been more convenient to review if page numbers and line numbers had been included, though. As this was not the case, my specific comments below are not assigned to individual lines, but rather to the respective paragraphs.

8Specific comments:

Please check your citations:

- The first mention of simultaneous expression of rTCA and CBB in *Riftia* symbionts inside the same host should be cited: DOI: 10.1126/science.1132913. This paper (from 2007) also addressed the influence of varying energy regimes on the expression of these two pathways. To say ‘...we have a limited understanding on whether these two pathways are constitutively active, or whether their activity relates to environmental conditions’, as stated in the introduction by Mitchell et al., is therefore not quite accurate. Same is true for the statements ‘the physiology of the symbiont, and ... are still largely unclear’ (discussion), and ‘it is unlikely that this pathway is completely restricted to a low oxygen regime’ (referring to the rTCA in the discussion – indeed, that the rTCA is simultaneously expressed with CBB, and not only under anoxic conditions, is not new).
- Might be worth mentioning that a putative role of hydrogenases in *Endoriftia*’s sulfur metabolism, as suggested by Mitchell et al., 2019, was also addressed in this study: <https://doi.org/10.7554/eLife.58371> (your reference 84).

Methods:

- Overall, the methods used are suitable to answer the research question. Statistical tests are appropriate and probability values are accurately described. Error bars appear only once (as far as I could see: in Figure S1) and should be described in the figure legend.
- In the description of the incubations, please add temperature and pressure details (even if these are included in the cited reference, it is worth mentioning them briefly, particularly since you are referring to ‘in situ pressures and temperatures’ in the discussion). On this note: Specific influence of pressure and temperature on gene expression is actually not discussed at all – why not?
- Please spell out CPM, PCA, MA on first use.
- Carbon incorporation: To which extent can this method elucidate the respective proportion of rTCA and CBB to overall carbon fixation under the analyzed conditions (i.e., is it a quantitative analysis)? Or can it only prove that both pathways are contributing (i.e., is it only a qualitative assessment)? Are conclusions on which carbon fixation pathway is connected to which conditions only based on the transcriptome analyses, or also on the carbon incorporation experiments? Please comment on this important question in the methods or results.
- What do you mean by ‘sampling occurs at the community level’ (section differential expression of key genes...)?

Discussion:

- Your sentence ‘However, the $\delta^{13}\text{C}$ values indicate a mixed carbon fixation strategy and genes for the rTCA show a higher or equal level of expression and protein abundance under most conditions’ – what do you mean? Higher or equal expression compared to what? Are you comparing expression levels of CBB enzymes vs rTCA enzymes? What would that tell you (beyond that both pathways are expressed)? As you are probably aware, RNA transcript abundance (even protein abundance) does not directly correspond to metabolic activity, and can therefore not actually reveal how ‘much the pathway is used’. Instead, comparisons should always be made across different conditions for the same enzymes.

Conclusion:

- You mention 'the possibility that each pathway is making differential contributions to biosynthesis'. While this has been suspected for a long time, your study could actually provide some more detailed answers: Which contributions would this be? Can you elaborate?

Formalities:

- Last sentence of the introduction ('Finally, the rates...'): very long complicated sentence. consider revising.
- Figure 1: Quite busy, can you make it a bit clearer?

Author Rebuttal to Initial comments

Response to referees

We thank the reviewers for their consideration and thoughtful critiques to and questions for this manuscript. It is our belief that all concerns are addressable. Below are our responses, and when applicable, line numbers in the manuscript where we incorporated changes to address these issues.

Reviewer 1:

Comments to the Author: The manuscript of Mitchell et al is an interesting study where Riftia tube worms and their microbial symbionts were subjected to high pressure experiments under different geochemical conditions to test the activity of two different carbon fixation pathways that are encoded by the symbionts genome. They also measured bulk DIC assimilation rates using ^{13}C bicarbonate into the host tissue (but it is unclear whether this is the microbial C-fixation rate, see below). As a general comment: there were no line numbers making it difficult to provide comments for specific portions of text and so I tried to be as specific as possible but without the line numbers my comments are relegated to being a bit more general in nature.

Reply: Line numbers were added.

Study summary: It's a really impressive experimental dataset with a high amount of replication and controls, and could meet the high mark for publication in Nature Microbiology I think. But

10there are many points that require clarification before publication could be considered (whether or not DNase treatment was applied, how long the worms were in the aquarium before the experiments were done, see below). Hopefully, the authors can address these points.

Overall comments:

I think that the authors actually are under utilizing their dataset, and should do some additional analyses to directly correlate gene expression data with carbon fixation rates under different conditions. This connection was not made as strong as it could be in the manuscript. For example, which genes correlate with higher/lower C-fixation rates? Do some genes increase expression under particular conditions that coincide with higher C-fixation? Were C-fixation rates lower when rTCA expression dominated, and was C-fixation higher when CBB expression dominated? This would go a long way to show which C-fixation pathway was more important and under which conditions. As far as I could tell, the authors primarily correlated the genes with one another (which is really interesting), but they don't correlate genes/pathways with C-fixation rates under different conditions. I think if they can do this it would help a lot to show which pathway is more important than the other under different conditions.

Reply: Thank for raising that point. To address the lack of clarity on how gene expression is linked to carbon fixation rates across these conditions, we added carbon fixation rate data to the Figure 2 panel so it is easier to see. In addition, we added a network-condition correlation analysis (Methods lines 402-420; Results lines 627-649). We believe this analysis addresses the main concern: What is the most relevant response to sulfide/oxygen limitation in this organism?

Initially, we had many panels showing pairwise comparisons allying carbon fixation rates to each of the ten experimental conditions. This was extremely onerous, and it was impractical to discern the patterns and overall relationships.

For example, the reader can see in Fig. 2B that under energy limiting conditions (ones in which we indeed saw lower carbon fixation rates), the gene expression patterns of *cbbM* and *acIA* did not show a universally consistent pattern, with the CBB sometimes also showing a relative increase in expression in two comparisons where we know that the carbon fixation rates were low. However, the co-expression analyses, especially with the suggested edits, provide a robust way to understand gene expression patterns across ten different conditions, and then link those to carbon fixation rates.

Below I have some more specific comments:

Abstract: Is it true that "most" organisms contain only one pathway? There are a lot of microbes out there and many of them have two pathways. Is there a citation to support this statement of fact? I think that the authors are probably correct, but still would be nice to have a citation to support this strong statement.

Reply: We thank the reviewer for this suggestion. There are two citations that we believe are

most relevant to this work: Berg et al. 2011 (<https://doi.org/10.1128/AEM.02473-10>) and Garritano et al. 2022 (<https://doi.org/10.1093/pnasnexus/pgac226>). In particular, Garritano et al. examined 52,515 metagenome assembled genomes (MAGs) and found 1,007 MAGs had one of the five carbon fixation pathways. Of those, only 23 MAGs had more than one pathway. Of course, that does not mean that they are functional, but it does indicate that dual pathways are indeed rare.

To minimize confusion, we also added this text in the introduction on **lines 136-137** that states that "most autotrophs are thought to possess one carbon fixation pathway (Berg et al. 2011; Garritano et al, 2022). We have added these citations.

In the second to last sentence of the abstract (and throughout the text), it is unclear what you mean "allied to".

Reply: Allied refers to genes that share first neighbors to one another: see text in manuscript, **lines 588-590** for a description of the biological relevance of first neighbors. On **line 590**, we added the phrase used in the literature, "guilt by association" to underscore this concept. On **line 772**, we added shared first neighbors (allied to) language to make it clear what we are referring to.

Introduction first paragraph: I think there are now six carbon fixation pathways known including the novel roTCA cycle (Steffens et al 2021 Nature).

Reply: Sorry for any confusion. The phrasing is “five other pathways” in addition to the CBB, so there are six “known” pathways and three other candidate pathways that we know of: roRTCA, reductive glycine pathway, and the reductive hexulose-phosphate pathway (RHP), which our citations cover. We changed the roRTCA citation to the Steffens paper, as suggested. (line 136).

Introduction last paragraph: it is unclear what you mean by "load balancing".

Reply: Thanks for the suggestion. We have added the following text to **lines 210-216**: “Here we use the phrase “load balancing” to refer to our hypothesis that these two pathways work in parallel to alleviate substrate limitations that might result from operating with only one pathway. One example is if nucleotide synthesis from ribose-5-phosphate originates principally from carbon fixed by the CBB cycle, and fatty acids are synthesized from acetyl-CoA derived from the rTCA cycle, the demand for acetyl-CoA from glyceraldehyde-3-phosphate (GAP) produced by the CBB cycle would be reduced.”

Methods: The worms were collected from a deep sea vent nearly 8 years ago, and it is stated they were placed into a high pressure aquarium afterwards. The authors need to clearly state how long the worms were in the aquarium before the experiments in the manuscript were conducted. Were the experiments in the manuscript done soon after collection or were the worms in the aquarium for several years prior to the experiments? It looks like the experiments were done on the ship and therefore soon after collection, but this was not entirely clear to me and should be stated in the methods.

Reply: We added the experimental hours to Table S1 in supplementary figures, and we added “Worms were given 12 to 24 hours to acclimate in aquaria before experimental conditions were started” in the methods section, **lines 259-260**.

Please provide full details on how the carbon fixation measurements were done (currently there is only a reference to Mitchell et al 2019 but this is insufficient, all key details should be provided here).

Reply: We added these details to the Supplementary Note 1 and referenced them in the methods section line 315.

How do you differentiate between the carbon fixed by the microbes, and the carbon that is transferred to the host tissue?

for raising this important point, which needs to be clarified. First, we are assuming the reviewer is referring to host carboxylation reactions that may occur in the plume, whereby CO₂ is incorporated via host pyruvate into a 4C organic acid (succinate and malate) and then transferred to the trophosome whereby it is either decarboxylated to CO₂ or transferred directly into the rTCA (or TCA) via TRAP transporters (although evidence for the latter is lacking). However, under CO₂ replete conditions, it has been shown that the vast majority of the carbon being fed into the symbiont's metabolism is in an inorganic form (Felbeck et al. 2004). That being stated, we did think about these organic acids that may be circulating in the blood, which would look like carbon fixed by the symbiont but are actually the result of a carboxylation reaction by the host. In our methods, we lyophilized the trophosome samples, took the dry weight, rinsed with acid, rinsed with DI water, and then dried them in an oven before grinding the tissues to be analyzed on an elemental analyzer that looks at ¹³C in the organic carbon portion of the sample. The acid removes any exogenous ¹³C from an inorganic source, whereas the rinsing steps removed any exogenous organic acids that may have been in the blood and/or coelomic fluid. Therefore, the carbon incorporation reported in this paper is from organic carbon that was either incorporated into the host tissue, or in the symbionts themselves. Keep in mind that host carboxylation reactions would only contribute to net carbon fixation if these organic acids were directly transferred to the symbionts and enter the rTCA cycle; if they were decarboxylated in the trophosome, this would not contribute to any of the organic carbon found in these tissues. While carboxylation reactions are common in animals, none can actually use these to provide a net gain of organic carbon for growth. If the hypothesis is correct that some of these are transferred directly to the symbiont's metabolism, this would still only be a very small portion of the total organic carbon incorporated, and it would still be dependent on the symbionts' metabolism, running the rTCA cycle, which is dependent on ATP and reducing equivalents, requiring redox reactions (e.g., sulfide oxidation) to run. Given that these host carboxylation reactions are independent of the symbiont's metabolism, the experiments that had no external sulfide supplied had net carbon incorporation values that were very small or zero, indicating that any host

carboxylation reactions are negligible to the overall net carbon incorporation rates. On lines 457-462, this was added to clarify: “Treatments that saw some minimal carbon incorporation without the addition of sulfide in the aquaria were likely utilizing elemental sulfur stores for autotrophy, and/or these were due to host carboxylation reactions.”

In addition, the data herein show the ^{13}C incorporation rates from *Riftia* worms that were provided with sulfide (the electron donor used by the symbionts to support carbon fixation), as well as from *Riftia* worms that were maintained in sulfide-free water. Those worms maintained in the absence of sulfide did not show any appreciable ^{13}C -incorporation. There is some evidence that the host has an initial carboxylation reaction in the plume, whereby CO_2 is incorporated via host pyruvate into a 4C organic acid (succinate and malate) and then transferred to the trophosome whereby it is either decarboxylated to CO_2 or transferred directly into the rTCA (or TCA) via TRAP transporters. If this is the case, any ^{13}C being incorporated via such carboxylations should be represented in the *Riftia* maintained without sulfide. Notably, these incorporation rates are substantially and significantly lower than the incorporation rates observed when the symbionts are provided with sulfide. To make this point clear to the reader, we have included the following sentence into the methods at lines 301-313: “The ^{13}C incorporation rates herein represent the net carbon fixation attributable to symbionts’ sulfide-dependent chemoautotrophic carbon fixation. Though there is evidence that the tubeworm can carboxylate pyruvate to a 4-carbon organic acid (such as succinate or malate)^{33,34}, those rates are insufficient to support net growth. Moreover, they would not likely be stimulated by the provision of sulfide, because only the symbionts can use that as an electron donor. Thus, any carbon incorporation due to host carboxylation reactions is represented by the ^{13}C incorporation rates measured in the absence of sulfide (i.e., the no sulfide conditions). Finally, to minimize carryover of inorganic carbon, all lyophilized samples were bathed in a 0.1N HCl solution for approximately ten minutes, rinsed repeatedly in deionized water, then dried in a gravimetric oven before being ground for elemental and isotopic analyses.”

Did you sample the microbes and the host tissue separately and make separate measurements? Currently it looks like you only sampled the host tissue, but the host does not fix carbon and so this is not a measurement of carbon fixation its a

measurement of how much of the fixed carbon from the symbiont was transferred to the host. Please clarify what exactly was measured (microbial biomass, or host biomass).

Reply: Yes, we sampled the symbiont-containing tissues (trophosome), as well as host tissues devoid of symbionts (gill and skin). The carbon incorporation values reported were from this trophosome tissue and represent the net carbon fixation due to autotrophic symbiont metabolism. Manuscript sentences on lines 297-299 addresses this point. In addition, on lines 300-301, we added “The trophosome tissue is ~24% symbionts by volume³² and is the vascularized organ that contains the specialized host cells that house the symbionts.”

How was DNA removed from the RNA extracts prior to sequencing? There is no DNase step reported in the methods, is it true that no DNase treatment was done? In that case, the transcriptomes are surely contaminated with DNA and cannot be published.

Reply: Sorry for the confusion. Yes, a DNase treatment was done, but it was done as part of the Microbial 'Omics Core (MOC) at the Broad institute (Cambridge, MA) protocol, which is outlined in the supplemental protocol of Shishkin et al. 2016. We added “DNA” to line 331 in the methods, to make this clear.

Results:

Please report in the text what the $\delta^{13}\text{C}$ of the Riftia biomass was after the incubations, currently it is only written that it was "far heavier" than the natural abundance. How much heavier? Or is that the "stable isotope ratios" of 40 and 181? Are those $\delta^{13}\text{C}$ values?

Reply: Yes, those are $\delta^{13}\text{C}$ values. We changed the text to clarify (lines 413-414)

It is interesting that you see over expression of cbbM with lower O₂, and higher expression at high O₂. This shows that the prevailing narrative that the CBB cycle is more active at high O₂ is not always true. I think this interesting finding could be highlighted more in the discussion (but it contradicts some statements in the conclusions, see below). Can you conclude why the CBB cycle is responding differently to low O₂ here than in other organisms?

Reply: Sorry for the confusion, we did not see overexpression of the CbbM gene under low O₂ conditions, as stated in **lines 508-510**. Rather, as in Fig. 2b, we see a decrease in expression of CbbM at low oxygen conditions (compared to high oxygen conditions; o_{vs}O). However, only one of these comparisons were significant.

How does the rDSR gene expression change with sulfide concentrations? And, how do the dissimilatory nitrate reductases change with nitrate concentrations? Do they should similar patterns to either the CBB or rTCA expression? It is written that they "share neighbors" with the rTCA. But this does not inform the reader whether they go up or down together or if they are decoupled, and it does not inform whether this is different between treatments.

Reply: Yes, we definitely want to make it clear that sharing neighbors means that there is co-regulation occurring. We added text on **lines 575-582** that hopefully makes this clearer. "Many relevant metabolic genes generally shared the same patterns of expression as the hub genes and shared neighbors. For example, the expression of genes of the rDSR (involved in sulfide oxidation) were relatively decreased under limiting conditions for oxygen and sulfide (similar to their hub neighbors PetB, *rpe*, *SqrA*, and *FccA*). The genes involved in denitrification (NorCB, *nosZ*, and *NirS*) had higher levels of expression under sulfide limitation (similar to their hub neighbors in the rTCA). For a complete list of genes, their DE patterns, and network relationships see GEO series accession #GSE249345."

In the absence of 15N isotope incorporation measurements, it is unclear to me how the authors can conclude that narGHI is being used for assimilation of nitrate (and not dissimilatory nitrate reduction).

Reply: We agree that more experimental evidence is needed to make that conclusion. However, we do feel that the interactions within the network suggest this is the case. We added the following text to explain our rationale on **lines 600-605** and believe that the language "these data support a model" is not an overstatement or conclusory: "However, in the network, the NarGHI genes shares first neighbors with GOGAT (glutamate synthase, which is a key assimilatory enzyme), and the Nap genes share first neighbors with *nosZ*, NorCB and *NirS*, which function in the dissimilatory reduction of nitrite to N₂. Therefore these data support a model whereby the Nar complex is operating in an assimilatory process and the Nap complex is being using

for dissimilatory nitrate utilization in Endoriftia (see Fig. 4a and 5).”

Discussion:

How do the rates of C fixation correlate with the expression of rTCA or CBB under the different treatments? Apologies if I missed it, but I couldn't find any discussion of this.

Reply: We added the carbon fixation rate measurements to Fig. 2, so it is easier for the reader to see. In addition, we added the following text to the discussion on lines 670-682” “Under conditions where carbon fixation was limited due to lower dissolved $\Sigma\text{H}_2\text{S}$ and O_2 concentrations, the network analysis revealed genes and pathways that are biological responses to these limiting environmental conditions (see Figs. 6a-d). Strikingly, sulfide oxidation genes and CBB genes were not among those responding to limiting conditions. Although many of these genes are DE under substrate limitation and appear as well-connected genes in the network (with some of them being hub genes), they were not found to be significantly correlated to these substrate limitations, whereas genes of the rTCA and the Hyd1e were. This suggests that the rTCA and Hyd1e are fundamentally important to the amelioration of energy limitation for the endosymbiont's metabolism. Moreover, many of the other genes involved in carbon metabolism that were significantly DE for these substrate limitations encode for bidirectional enzymes (FrdCAB, PFOR(3), PFOR-like, and KorAB(3)). However, these genes also have isoforms elsewhere in the genome and could be play an important role in the metabolic response due to their particular redox requirements and may represent a genetic redundancy that could have furthered niche expansion.”

The statement in the conclusions that "Surprisingly ... CBB (broadly speaking) allied to more aerobic processes and rTCA allied to anaerobic processes." , is a bit of a surprise after reading the discussion and results. That conclusion did not come out clearly in the main text, and seems a bit contradictory with some results (higher cbbM expression at low O_2 , higher expression of rTCA at low sulfide, see above). Can you please clarify this point?

Reply: Again, sorry for the confusion, the CBB did not show higher expression under low oxygen conditions. The aerobic processes (albeit not exclusively) we are referring to are the ETC genes (PetABC and CBB_3) and the F-type ATPase, which share neighbors with the CBB. Whereas the rTCA shares first neighbors

18with denitrification genes, an MBX complex, V-type ATPases, and even the Hdr-Fix genes within the rTCA operon are all associated with more anaerobic metabolisms. We used the term “broadly speaking,” since there are exceptions (with the symbiont likely being one of them).

"This study has provided the first in depth look at an organism that simultaneously uses two pathways whose substrates and endproducts are allied to different metabolic processes,"

Are you sure you are the first? Giovanelli et al (eLife) looked at co-occurring carbon fixation pathways (rTCA and WLP) in *Thermovibrio ammoniensis*. Moreover, some methanogens have co-occurring WLP and rTCA cycles for carbon fixation.

Reply: We do not claim to be the first to look at the phenomenon of dual carbon fixation pathways in an organism. There are several previous studies, which we cite, that have incorporated various combinations of genomic, transcriptomic, or proteomic data of freshly recovered organisms. Other authors have taken a completely *in silico* approach, using modeling to see how two pathways would interact with each other in one organism (Cheng et al. 2019 and Sumi, T., & Harada, K. 2021). However, we posit that this is the most comprehensive study to date, as we studied how metabolic rates and mRNA transcription co-vary across the range of environmentally-relevant conditions found at deep sea vents.

We are also excited that there is a preprint from Berg et al. that examines how *Ammonifex degensii* use the Wood–Ljungdahl and the CBB pathways. This study also used ¹³C inorganic carbon at environmentally-relevant conditions.

This being said, these dual pathways and/or hybrid pathways are very much an active area of research in microbial physiology, as well as synthetic biology. We believe the work presented herein furthers this research significantly, as it is the first study to utilize a co-expression analysis, under verifiable steady state conditions, which linked environmental conditions with a complete suite of metabolic processes related to each pathway. This study presents a powerful tool that has been underutilized in this area of research, and it offers abundant data that can pave the way for more hypothesis-driven investigations. Furthermore, it offers crucial insights for upcoming endeavors in metabolic and kinetic modeling.

Reviewer 2:

Comments to the Author:

This paper describes RNA-level co-expression patterns in Endoriftia symbionts of Riftia tubeworms, based on the impressive pressure incubations onboard, immediately following animal retrieval from the deep-sea hydrothermal vents. The authors focus on a key question regarding the ability of these organisms to fix copious amounts of carbon, based on the potential use of both the CBB and rTCA cycles by the symbionts, asking how environmental conditions affect these processes. Using coexpression network analyses, the authors identified key modules and hub genes in Endoriftia metabolism.

Despite the impressive experimental setup, the findings are not highly novel, as they are often confirmatory of previous work testing the expression of Endoriftia proteins in S-rich and S-depleted specimens, and showing the differential expression of CBB and rTCA enzymes (Markert 2007, Leonard et al 2021, Hinzke et al 2021). While this work tests a larger array of conditions, the conclusions mainly highlight the difference between the high and low sulfide conditions. The effects of other conditions were not discussed in detail – for example, Figure 4 is mentioned only briefly.

Another fascinating aspect of this study is the measurements of carbon uptake rates, as well as those of the use of other resources. Yet, the high uptake rates from such experiments were already reported previously by Mitchell et al. 2019, thus the novelty here is also limited.

There is a possible issue of pseudoreplication, given that the experiments were conducted on individuals placed in a single container (is it indeed the case?). This may lead to unknown bottle effects. The authors should at least address this issue, and explain why the results are valid.

Additional topics that require attention include the effects of radial variability in expression patterns (Hinzke et al., 2021), as well as the potential adaptability at the sub-species level (Polzin et al., 2019).

Reply: We thank the reviewer for their feedback. We will address some of the broad feedback here and the rest in the next section. First and foremost, we would

20like to state that it is our belief that these data and the resulting analyses provide novel insights into the functioning of these symbionts. Some of the findings, of course, are consistent with previous assertions, but we do not believe this takes away from this work's importance. This work also significantly adds to the body of knowledge regarding the symbiont's metabolism, as well as a metabolism that operates two carbon fixation pathways. It is completely novel for the following reasons:

- a) There was a large breadth of information used (carbon fixation rates, geochemical conditions, gene expression, and finally, a co-expression analysis).
- b) The co-expression analysis revealed surprises that are relevant to a much broader field of research. Namely, the way the network is partitioned around these two pathways underscores the importance of both for metabolic functioning. These data also generated a plethora of metabolic linkages that were simply unknown before, providing a wealth of information for further hypothesis driven experimentation.
- c) These data revealed that a Group 1e hydrogenase likely plays a pivotal role in the metabolic response of sulfide and oxygen limitation. Given that we already know that the symbiont does not take up hydrogen and use it for carbon fixation, this begs the question of what it is doing.
- d) There are many genes of currently unclear functions in the symbiont (MBX complex, both of the hydrogenases, the V-type ATPases, the Hdr-Flx, the QFR/SQR frdCAB, not to mention the genes of less clear annotations and the hypotheticals) that this work links to other systems in the symbiont's metabolism. It is our belief that this body of work will be very useful to researchers that study these systems.

Other comments:

Introduction:

Please fix Hdr-flx to "NADH dehydrogenase/heterodisulfide reductase (Hdr-Flx)"

Expand the PPP acronym.

Reply: These are good points, and we thank the reviewer. We expanded these abbreviations on lines 195-196 & 474.

Methods:

Can you comment on the potential bottle effects? If I understand correctly, only a single tank was assigned per condition. Wouldn't place individuals in the same tank account as pseudoreplication? For example, see here - [10.1371/journal.pbio.2005282](https://doi.org/10.1371/journal.pbio.2005282).

Reply: From the paper cited above: "In some experiments, unrelated animals may be (1) individually randomised to treatment groups and then housed together by their assigned treatment group or (2) individually randomised to cages and then cages are randomised to treatment groups. In both cases animals, and not cages, can be considered the EU because the animals are independently assigned to the treatment groups (first criterion is met), but the second and third criteria must also be met for animals to be considered the EU." The second and third criteria are treatment applied independently and the animals do not influence each other.

The reviewer is correct in that each condition was done in the same vessel with multiple tubeworms and raises questions as to whether the latter two criteria were met. In this particular case, we believe our experimental design provides the sufficient statistical robustness. We brought the reviewer's concern to the attention of Dr. Steve Worthington, who is the Director of Data Science Services and the Lead Data Scientist at Harvard University. He was not concerned, and gave the following reasons:

- 1. These worms are found together in clumps, so their natural state is to be close together. Separating them into different vessels could potentially introduce another, even more challenging bias that would be difficult to control.**
- 2. We are looking at a symbiont population within the worms and not the worms per se. Thus, the "biological unit" in these experiments is the independent sampling of the symbiont populations within each worm. The symbionts are completely separated from the symbionts of another tubeworm while in the vessel, yet the worm they inhabit has the same environmental conditions.**
- 3. Dr. Worthington did acknowledge there may be a spillover effect. The worms in one vessel could affect the other worms by altering the geochemical conditions in the vessels. We have decades of experience in mitigating these effects and did so in these experiments by A) doing these experiments in flow-through reactors that are constantly being flushed with fresh vent-like seawater; and B) controlling the dissolved geochemical concentrations to ensure that none of the substrates unintentionally became limiting. We monitor both flow rate and dissolved chemical concentrations to ensure that we achieve steady state before commencing any experimentation.**

22Overall, he felt confident that our methods did not compromise the results in a way that wasn't accounted for.

Oxygen levels in 2014 experiments appear to be very high – how do they compare to the in-situ values? Can you comment on the potential effects of these high values?

Reply: Sorry for the confusion, we added the phrase “intake values” to the column header in that table to make it clearer. These values are the oxygen concentrations that were measured from the equilibration columns feeding the vessels. *Riftia* rapidly and substantially draw down oxygen in the aquaria, which results in an “apparent” oxygen concentration around the worms of ~100 μM . That is representative of *in situ* oxygen concentrations. In the oxygen limitation treatments, we exposed them to markedly less dissolved oxygen, such that they often pulled the concentrations to zero, which is by definition limiting.

Animal sectioning – How the trophosome subsections for RNA extractions were chosen? Could you comment on how your work can cope with the radial variability – e.g., <https://doi.org/10.7554/eLife.58371>. Could this explain the outliers you found in the treatments?

Reply: Work by Bright et al. 2000 and Bright and Sorgo, 2003 provide evidence that symbionts and the host cells that they occupy have a coordinated cell cycle, which also correlates with symbiont sizes. The smaller rod-shaped symbionts undergo cell divisions, along with the host cell bacteriocytes in the interior of the trophosome lobule. As you move away from the central blood vessel, these host cells and the bacteria within them increase in size, become coccoid, and stop dividing, with an eventual degradation in the lobule periphery. The blood flow, which contains the substrates needed for their metabolism, flows from the periphery to the central blood vessel, and it has been suggested that this flow direction may cause differences in substrate availability within the trophosome lobule, possibly also contributing to cell size differentiation. Hinzke et al. 2021 undertook the heroic and challenging task of looking at differences in protein expression among not only the different symbiont size classes within the trophosome, but also sulfur-depleted and sulfur-replete symbionts. We think the term “sub

23populations” can be a bit confusing. It was shown by Polzin et al. 2019 that among the 16S rRNA phylotype of *Endoriftia*, there are different strains, with one strain being very dominant (“STR-1”). However, the Hinzke et al. 2021 paper is not referring to these different genotypes (they did not look at that); they were looking at the different symbiont size classes described above. It is important to point out here that these size classes are thought to be a part of the same cell cycle, with small symbionts being the “stem cells” and later becoming the larger symbionts.

The Hinzke et al. 2021 data revealed the fine scale differences in protein composition and isotope ratio within an animal but do not speak to the influence of environmental conditions on net chemoautotrophic functioning. For example, the authors did not see significant differences between proteins involved in rTCA and CBB between these two symbiont groups. They did see a difference in relative stable carbon isotope fingerprint values between fractions of sulfur-depleted symbionts, which had relatively heavier isotope signatures, indicative of a greater reliance on the rTCA under low energy conditions. The authors also found differences in relative protein abundances between the different size fractions of the symbionts. These are insightful findings, but these authors were using worms that were collected from the study site, and the time from collection to being on board can be hours, with periods of hypoxia in the biobox during the ascent. In addition, the authors used the color of trophosome as a proxy for historical environmental conditions, but we know from our extensive high-pressure experimental studies that trophosome color is not a well-constrained proxy for the history of sulfide exposure and not at all a proxy for symbiont metabolic activity. Thus, the data within Hinzke et al. 2021 provide remarkable insights into the variation within an animal but do not represent the influence of environmental conditions on net chemoautotrophic function, or the metabolic response to substrate limitations.

The work presented in this manuscript does not differentiate between size class, or the possibility of different genotypes, though the majority of host associated symbiont populations are “STR-1” dominated. We did, however, use bowtie2 in “very sensitive local mode” and RSEM against the reference genome (which is the dominant strain). This mode is specifically designed to capture genes with possible strain level variations and did not capture any such genes (Langmead and Salzberg 2012). That being said, small strain level differences, if present, are unlikely to significantly shift these gross level responses for the reasons described above.

Regarding the “sub populations” of symbionts of different sizes that appear to have different metabolic states, we designed our sampling protocol to minimize such biases. Specifically, we randomly picked (or to be more accurate, we haphazardly picked) the sampling locale from each trophosome. Moreover, each specific trophosome sample was large enough in mass that all symbiont size classes were likely represented. Thus, our data presented here represents the population as a whole.

24Removal of the overrepresented reads - Wouldn't this affect the data? For example, some genes in tubeworm symbionts, such as those that encode porins and sulfur globules, could have exceptional expression values.

Reply: Over-represented sequences in RNA-seq reads are not typically expected for protein-coding transcripts. For such over-representation to occur, there must be an excess of reads beginning at a particular position in a transcript. In a typical protein coding transcript, given a relatively larger length of a transcript relative to a library fragment, there are multiple possible start positions, with variation due to the length of the underlying transcript, as well as the sequence fragment from which the reads were generated. Over-represented reads typically originate from rRNAs, erroneously sequenced adapter fragments, or other transcriptional noise, and running nucleotide BLAST searches for a subset of those sequences confirmed this expectation. Removing over-represented reads and putative rRNA reads that mapped to the SILVA database is a way to a) remove reads that do not originate from protein-coding loci, and b) for the readers, provide a more realistic picture of the actual number of reads that went into expression estimation, and thus whether the amount of data per sample is sufficient for the analyses we undertook.

Fix reference placement: database (release 138, 31)

Reply: We changed this to: “database³⁸ (release 138)” in line 356

What was the purpose of filtering out the rRNA sequences, given that they don't play a role in the comparative dataset?

Reply: We filtered out rRNA sequences during the library prep stage, as well as afterwards *in silico*, because they tend to drown out and dominate the datasets. They also tend to be constant across different conditions, so they don't really provide meaningful information about DE gene expression. A depletion of the rRNA during library prep allows for deeper sequencing of the protein coding genes, thus increasing depth of coverage and the ability to detect low abundance transcripts.

25Results:

In “Gross patterns in symbiont gene expression” Please rephrase “sufficiently abundant”.

Reply: We rephrased to “Overall patterns in symbiont gene expression” since “sufficiently abundant” does not cover everything that we discuss in that paragraph (line 464).

In “Evidence of energy conservation linked to the rTCA”, please provide evidence for the assumption that the rTCA genes form a functional operon.

Reply: Lines of evidence of a functional operon:

1. This gene cluster is well conserved (Rubin-Blum et al. 2019)
2. These genes are co-transcribed (see Extended Data Fig 3a).
3. Many of these genes have very short intergenic distances between them, with a few of them overlapping (see Extended Data Fig 3b).
4. We see continuous read coverage along the whole operon length(see Extended Data Fig 3b).
5. While Hdr-Flx and the transhydrogenase (PntAB) are not rTCA genes, the rest are part of this pathway, and they are interspersed along the length of this gene cluster, such that there is a functional relatedness in these genes.

Please check for continuous read coverage along the whole operon transcript.

Reply: We ran this analysis, and indeed saw read coverage along the length of this operon and is shown in the Extended Data Fig. 3b.

In “First neighbor analysis reveals a bi-modal metabolism” – “A first neighbor analysis was performed on the major metabolic pathways involved in energy metabolism” – energy or carbon?

Reply: We changed that to “energy and carbon” line 501

“Genes related to sulfide metabolism show a clear pattern whereby most sulfide oxidation systems (sox, rDSR, apr/sat, sqrA, and fccA)...” there is a mixture between genes, proteins and systems...

Reply: We changed the text to: “Genes related to sulfide metabolism show a clear pattern in which most sulfide oxidation genes (those in the sox and rDSR pathways, as well as *aprA*, *aprB*, *sat*, *SqrA* and *FccA*) share first neighbors with the CBB and not the rTCA. The exception to this was seen in genes for sulfur globule proteins (Sgp) and genes for the membrane bound sulfite-oxidizing enzyme (Soe), both of which only have first neighbors with the rTCA” (lines 605-625).

Discussion:

“Surprisingly, these analyses revealed that the rTCA and CBB carbon fixation pathways form metabolic nodes that are coupled to other distinct metabolic processes, which may explain their ability to support such high growth rates in Riftia” – While not done in the same systematic manner as the network analysis employed here, previous studies appear to show some of these, and also potentially contradicting co-expression patterns (e.g., cell size-dependent expression patterns seem to indicate co-expression of RubisCO and denitrification in Hinzke et al 2021).

Reply: See our reply above in response to whether this work is novel and contributes to new insights. In response to the Hinzke et al. 2021 dataset being contradictory:

These two datasets are measuring two very different things. Hinzke et al. 2021 looked at differences in protein abundances between different size classes. They found that genes for carbon fixation from both rTCA and CBB have higher relative abundances in larger symbionts, as well as genes for denitrification, NiFe1e, and sox. This is not in contradiction to our results because these data are not measuring a metabolic response to a limitation. It is our belief that the Hinzke et al. 2021 paper is the best evidence, to date, that both pathways are operating simultaneously. Whereas the data presented in this manuscript represent the mean metabolic response of the whole population under varying conditions, revealing co-expression patterns that provide a window into how two simultaneous pathways may operate.

Figure 1:

I think it is worth moving column 5 to column 3, to keep similar treatments together.

Reply: We thank the reviewer for the suggestion and changed these columns in this figure.

Please fix (_ indicates seawater only).

Reply: We changed it to w – bottom water.

In the legend, please expand DE to differential expression.

Reply: Done.

Figure 2b: Panels B, D – there are additional clusters of upregulated genes – which genes are there?

Reply: Yes, those are flagella genes and chemotaxis genes mostly. They form their own module (gre60) and share no first neighbors with any of the genes involved with metabolism. We chose not to discuss these, as it would have been a departure from the focus on metabolism. However, we will be including the DE and the co-expression data tables so that everyone will have access to these data. There is a lot of information contained within these data that we think other researchers may find very useful. That being said, readers can look at the list of functions of note in the modules from Figure 3 to get a general sense of what might be upregulated/downregulated in 2b (now Extended Data Fig. 5b).

In the legend, please expand DE to differential expression. ETC?

Reply: We expanded DE, and ETC (electron transport chain) has been defined earlier in the text (line 566).

References styling is inconsistent.

Reply: This has been addressed.

Reviewer 3

Reviewer #3 (Remarks to the Author):

In their study, Mitchell et al. have investigated the two carbon fixation pathways in the hydrothermal vent tubeworm symbiont *Candidatus Endoriffia persephone*. Using

28incubation experiments in high pressure aquaria, carbon isotope labelling and transcriptomic analyses, the authors aimed to determine a) how different environmental conditions influence the expression of the two pathways – the Calvin-Benson-Bassam (CBB) cycle and the reductive TCA (rTCA) cycle – and b) whether these two CO₂ fixation pathways have distinct cellular functions, e.g., in providing metabolic precursors – and thus basically c) why the symbiont uses two CO₂ fixation pathways simultaneously.

The study provides some interesting new findings, namely that rTCA and CBB seem to be ‘allied’ to distinct sets of metabolic pathways, with hardly any overlap between these two hubs, as revealed by co-expression network analyses/first neighbor analysis. This method and its results are indeed innovative and enhance our current understanding of this model symbiont’s physiology. Other approaches and findings, such as the observation that rTCA enzymes are particularly abundant when ambient sulfide concentrations are low and energy is limited, confirm previous studies (see below) and are therefore valuable, but not particularly original. The results are of immediate interest to researchers in the field of hydrothermal vent microbiology and physiology, as they add another interesting piece to the puzzle of how gutless tubeworms and their sulfur-oxidizing bacterial symbionts have optimized their extraordinary association. To be interesting to a broader audience, the study would need to elaborate more on how the symbiosis benefits, specifically, from the observed metabolic nodes, one centered around rTCA, the other centered around CBB, (beyond the relatively general assumption of an unspecified ‘fitness advantage’). What shapes these ‘alliances’ between pathways? As it is, the manuscript provides a solid basis for in depth follow-up research – but does, regrettably, not go into further detail to explore and explain individual connections within the nodes.

The manuscript is well written (except for a few minor sloppy mistakes) and clearly comprehensible. It would have been more convenient to review if page numbers and line numbers had been included, though. As this was not the case, my specific comments below are not assigned to individual lines, but rather to the respective paragraphs.

Reply: We thank the reviewer for their feedback and agree that we have been careful in our conclusions.

With that in mind, we decided to include another analysis: a module-condition analysis that more directly links the network patterns with conditions. This analysis underscored the essentiality of both the rTCA and the Hyd1e to the metabolic response for sulfide limitation (and oxygen limitation with the Hyd1e as well). We think that these results are very different from showing a difference in

29genes or proteins between two different conditions (as in previous studies). While it validates previous researchers' hypotheses about the rTCA being important under energy limiting conditions, it also shows that the rTCA is one of the most important responses to sulfide limitation, that has never been demonstrated before—or even suggested.

In addition, this work shows the Hyd1e playing a pivotal role in this metabolism, in a hitherto unknown way. This is also quite new.

Unfortunately, the techniques needed to fully probe metabolic functioning are unavailable to us, given the difficulty of working with these organisms. They cannot be cultured. Nor do they remain viable after separation from their host for very long, and these techniques also introduce other biases. This unfortunately excludes many of the metabolic physiology experiments that could be used to fully explore individual components of their metabolism.

That being said, we believe that this work still represents one of the most in depth looks at a chemoautotrophic metabolism using dual pathways. While some of the findings did support previous researchers' findings (which is good), many of our findings are very new, and we think can be used by researchers for many more hypothesis-driven experiments.

Specific comments:

Please check your citations:

- The first mention of simultaneous expression of rTCA and CBB in Riftia symbionts inside the same host should be cited: DOI: 10.1126/science.1132913. This paper (from 2007) also addressed the influence of varying energy regimes on the expression of these two pathways. To say '...we have a limited understanding on whether these two pathways are constitutively active, or whether their activity relates to environmental conditions', as stated in the introduction by Mitchell et al., is therefore not quite accurate.

Reply: We added this reference and changed the wording in the introduction:

“That said, we have a limited understanding of whether these two pathways are constitutively active, and how this activity relates to environmental conditions. Moreover, we have no understanding of how these pathways integrate with other metabolic processes” lines 151-154.

Same is true for the statements 'the physiology of the symbiont, and ... are still largely unclear' (discussion), and 'it is unlikely that this pathway is completely restricted to a low

30oxygen regime' (referring to the rTCA in the discussion – indeed, that the rTCA is simultaneously expressed with CBB, and not only under anoxic conditions, is not new).

Reply: Respectfully, we believe the new and unexpected patterns of gene expression across these treatments underscores how little we actually know about the physiology of this endosymbiont. The discussion about oxygen is a case in point. The symbiont has and expresses multiple oxygen sensitive systems, many of which are co-expressed with the rTCA (also oxygen sensitive).

Might be worth mentioning that a putative role of hydrogenases in Endoriftia's sulfur metabolism, as suggested by Mitchell et al., 2019, was also addressed in this study: <https://doi.org/10.7554/eLife.58371> (your reference 84).

Reply: Hinzke et al. 2021 did not address the hydrogen metabolism per se; they saw that the hydrogenases were expressed and discussed what they could be doing (citing Mitchell et al. 2019). It seems a bit circular to cite that.

Methods:

Overall, the methods used are suitable to answer the research question. Statistical tests are appropriate and probability values are accurately described. Error bars appear only once (as far as I could see: in Figure S1) and should be described in the figure legend.

Reply: We thank the reviewer for catching that and have added it in.

In the description of the incubations, please add temperature and pressure details (even if these are included in the cited reference, it is worth mentioning them briefly, particularly since you are referring to 'in situ pressures and temperatures' in the discussion).

Reply: We added these to the methods on line 256-258.

On this note: Specific influence of pressure and temperature on gene expression is actually not discussed at all – why not?

Reply: We are not aware of a specific study that looks at gene expression changes at different temperatures and pressures in *Riftia* or their symbionts. However, based on our experience, the host tubeworm dies within a few hours at atmospheric pressures (our

observation and Childress et al. 1984). Also, the rates of carbon fixation are considerably lower than the rates found in the intact association (Scott et al. 1994, Felbeck and Jarchow 1998). In general, we know that pressure influences the gene expression in bacteria (Bartlett, D. H. 2001). In addition, when deep sea bacteria are incubated at *in situ* pressures, they show much higher productivity rates (Bianchi et al. 1999). These issues, in our minds, justified the development and use of the high-pressure systems for conducting these studies.

As far as temperature goes, previous studies have shown optimal metabolic rates at 25 °C. However, those were short term incubations, and while *Riftia* can be found at those temperatures, temperatures between 10-15 °C are more common. We chose to do our incubations at 15 °C and at 17 Mpa for these reasons.

Please spell out CPM, PCA, MA on first use.

Reply: We thank the reviewer for catching that. Done.

Carbon incorporation: To which extent can this method elucidate the respective proportion of rTCA and CBB to overall carbon fixation under the analyzed conditions (i.e., is it a quantitative analysis)? Or can it only prove that both pathways are contributing (i.e., is it only a qualitative assessment)? Are conclusions on which carbon fixation pathway is connected to which conditions only based on the transcriptome analyses, or also on the carbon incorporation experiments? Please comment on this important question in the methods or results.

Reply: These methods were not able to tease apart which pathway was contributing more to overall carbon fixation under these experimental conditions. These data show that under conditions where carbon fixation is low due to sulfide limitation, the rTCA shows a higher level of expression, etc. (So yes, it is qualitative.)

We added, “These data should be interpreted solely as a response in gene expression, not a quantitative representation of carbon fixation rates” (lines 516-517).

What do you mean by ‘sampling occurs at the community level’ (section differential expression of key genes...)?

Reply: This is a reference to what was found in the study we were citing. On a single cell level, mRNA abundance tends to be very poorly correlated with protein abundance. However, when looking at a larger population, after steady state has been reached, there is a much higher correlation such that the mean mRNA is a good predictor of mean protein abundance.

We changed the paragraph to “We also recognize that gene transcription does not necessarily correlate with protein abundances, though we also note that previous studies have found a much tighter correlation between gene transcription and protein abundances when sampling communities (i.e., when looking at population-level abundances) after environmental steady states have been reached (that is the case in this study)⁶⁰.”, on **lines 517-522**.

Discussion:

Your sentence ‘However, the $\delta^{13}\text{C}$ values indicate a mixed carbon fixation strategy and genes for the rTCA show a higher or equal level of expression and protein abundance under most conditions’ – what do you mean? Higher or equal expression compared to what? Are you comparing expression levels of CBB enzymes vs rTCA enzymes? What would that tell you (beyond that both pathways are expressed)? As you are probably aware, RNA transcript abundance (even protein abundance) does not directly correspond to metabolic activity, and can therefore not actually reveal how ‘much the pathway is used’. Instead, comparisons should always be made across different conditions for the same enzymes.

Reply: We changed this to “genes for both pathways are highly expressed” (**line 738**).

Conclusion:

You mention ‘the possibility that each pathway is making differential contributions to biosynthesis’. While this has been suspected for a long time, your study could actually provide some more detailed answers: Which contributions would this be? Can you elaborate?

Reply: The co-expression data set the stage for future studies that can address if and how each of these pathways is making differential contributions to biosynthesis. For example, these analyses link different systems to each pathway, e.g., glycogen metabolism linked to the rTCA, and future experiments could be designed to test these hypotheses. We do not think these data, as a stand-alone,

33can tell us what these differential contributions are. This is primarily because most of these treatments were done with a limitation of either oxygen, sulfide, or nitrogen. In other words, in many of these experiments, biomass production was depressed.

Formalities:

Last sentence of the introduction ('Finally, the rates...'): very long complicated sentence. consider revising.

Reply: We changed the last sentence to: "Finally, the distinct bi-modal distribution of other metabolic processes around these two functionally degenerate carbon fixation pathways underscores the importance of both and hints at a regulatory independence that may offer more protection to perturbations²⁷" (lines 236-239).

Figure 1: Quite busy, can you make it a bit clearer?

Reply: We simplified this figure, hopefully it is clearer now.

Decision Letter, first revision:

Message: Our ref: NMICROBIOL-23061497A

9th February 2024

Dear Dr. Girguis,

Thank you for your patience as we've prepared the guidelines for final submission of your Nature Microbiology manuscript, "Beyond the single pathway: co-expression analysis reveals distinct roles of dual carbon fixation in *Riftia pachyptila* symbionts" (NMICROBIOL-23061497A). Please carefully follow the step-by-step instructions provided in the attached file, and add a response in each row of the table to indicate the changes that you have

34made. Please also check and comment on any additional marked-up edits we have proposed within the text. Ensuring that each point is addressed will help to ensure that your revised manuscript can be swiftly handed over to our production team.

In recognition of the time and expertise our reviewers provide to Nature Microbiology's editorial process, we would like to formally acknowledge their contribution to the external peer review of your manuscript entitled "Beyond the single pathway: co-expression analysis reveals distinct roles of dual carbon fixation in *Riftia pachyptila* symbionts". For those reviewers who give their assent, we will be publishing their names alongside the published article.

Nature Microbiology offers a Transparent Peer Review option for new original research manuscripts submitted after December 1st, 2019. As part of this initiative, we encourage our authors to support increased transparency into the peer review process by agreeing to have the reviewer comments, author rebuttal letters, and editorial decision letters published as a Supplementary item. When you submit your final files please clearly state in your cover letter whether or not you would like to participate in this initiative. Please note that failure to state your preference will result in delays in accepting your manuscript for publication.

Cover suggestions

COVER ARTWORK: We welcome submissions of artwork for consideration for our cover. For more information, please see our guide for cover artwork.

Nature Microbiology has now transitioned to a unified Rights Collection system which will allow our Author Services team to quickly and easily collect the rights and permissions required to publish your work. Approximately 10 days after your paper is formally accepted, you will receive an email in providing you with a link to complete the grant of rights. If your paper is eligible for Open Access, our Author Services team will also be in touch regarding any additional information that may be required to arrange payment for your article.

Please note that *Nature Microbiology* is a Transformative Journal (TJ). Authors may publish their research with us through the traditional subscription access route or make their paper immediately open access through payment of an article-processing charge (APC). Authors will not be required to make a final decision about access to their article until it has been accepted. Find out more about Transformative Journals

Best regards,

Reviewer #1:

Remarks to the Author:

The authors have done a good job responding to my comments, and the new changes to the text and figures are appreciated and satisfied my comments and questions. With these new changes, I recommend the manuscript for publication.

Reviewer #2:

Remarks to the Author:

I thank the authors for the detailed response to the comments.

I kindly ask you to address the issue of experimental unit statistics in the methods part of the main text, not only in the rebuttal letter. Providing the statistical framework for this kind of work would benefit the community, particularly those who work on ex-situ host-microbe interactions.

36Reviewer #3:

Remarks to the Author:

The manuscript has indeed noticeably improved and benefited from careful revision. It is now more focused on the truly innovative aspects of this comprehensive study. Specifically, the additional module-condition analysis that more directly links the network patterns with external environmental conditions yields new insights that go beyond those from previous analyses. Most concerns raised by the reviewers were satisfyingly addressed, and the manuscript adjusted accordingly. Studying deep sea symbioses under in-situ conditions is technically extremely challenging, and the methods used here, i.e., incubations in high pressure aquaria, are elaborate and present the most advanced options available for these research questions. The results presented in this study are a diligently and meticulously collected, solid and very detailed data resource for future studies. As the authors also state in their response letter, their results "can be used by researchers for many more hypothesis-driven experiments". I absolutely agree. Still, to make the manuscript relevant and of general interest for a broader audience, I would have loved to see some of these potential follow-up questions – how, specifically, does the symbiosis benefit from this clustering of metabolic functions and what drives this co-expression? – addressed in this manuscript.

Some tiny additional comments:

Methods: You might want to spell out HOV (human occupied vehicle) for readers who are not familiar with the abbreviation. For the mRNA sampling, it might be worth specifying that tissue samples from individual worms represent biological replicates for the respective condition (and were analyzed separately). Did you take RNA samples from all 30 worm individuals? Cannot hurt to specify it here.

Results: line 531 "In our DE analyses, nearly all the genes in this cluster all increase or decrease..." remove the second "all" (redundant); line 599: "the NarGHI genes shares" - please correct (the NarGHI genes share)

Discussion: line 681: "these genes also have isoforms elsewhere in the genome and could be play an important role" - please correct

Author Rebuttal, first revision:

Thank you for all of your insightful comments and suggestions, we hope that the following edits addresses your remaining concerns.

Reviewer #1 (Remarks to the Author):

The authors have done a good job responding to my comments, and the new changes to the text and figures are appreciated and satisfied my comments and questions. With these new

37changes, I recommend the manuscript for publication.

Reviewer #2 (Remarks to the Author):

I thank the authors for the detailed response to the comments.

I kindly ask you to address the issue of experimental unit statistics in the methods part of the main text, not only in the rebuttal letter. Providing the statistical framework for this kind of work would benefit the community, particularly those who work on ex-situ host-microbe interactions.

Response: Thank you for the suggestion we added a paragraph in the methods section, line 405:

“It is worth noting that by incubating multiple *Riftia* in each vessel, the treatments were not applied independently, introducing a possible spillover effect. Given the difficulty with working with deep-sea organisms, as well as the lifestyle of these worms, we felt that this was the best method because 1) limited time at sea meant it was impractical to incubate 30 worms in separate aquaria; 2) aquaria seawater conditions were effectively steady state, thus minimizing spillover effects due to geochemistry; and 3) *Riftia* is found in very tightly packed clumps, and we believe that putting them together more closely mimics their *in situ* conditions. Finally, this study focuses on the symbiont population within each *Riftia*, wherein each worm already governs the conditions around the symbionts.”

Reviewer #3 (Remarks to the Author):

The manuscript has indeed noticeably improved and benefited from careful revision. It is now more focused on the truly innovative aspects of this comprehensive study. Specifically, the additional module-condition analysis that more directly links the network patterns with external environmental conditions yields new insights that go beyond those from previous analyses. Most concerns raised by the reviewers were satisfyingly addressed, and the manuscript

38adjusted accordingly. Studying deep sea symbioses under in-situ conditions is technically extremely challenging, and the methods used here, i.e., incubations in high pressure aquaria, are elaborate and present the most advanced options available for these research questions. The results presented in this study are a diligently and meticulously collected, solid and very detailed data resource for future studies. As the authors also state in their response letter, their results “can be used by researchers for many more hypothesis-driven experiments”. I absolutely agree. Still, to make the manuscript relevant and of general interest for a broader audience, I would have loved to see some of these potential follow-up questions – how, specifically, does the symbiosis benefit from this clustering of metabolic functions and what drives this co-expression? – addressed in this manuscript.

Response: We believe these three statements in the discussion section addresses this:

Line 354: This strategy may enable the host-symbiont association to have high productivity rates and maintain autotrophic poise in the stochastic vent environment.

Line 356: Finally, modularity in co-expression networks, such as seen in our data, is thought to emerge in organisms that encounter environmental stressors, where this reorganization of the network may increase robustness to perturbation.

Line 362: That said, we posit that this mode of autotrophy may represent a new carbon fixation modality that confers a fitness advantage in a highly dynamic environment where redox conditions are continuously changing.

Some tiny additional comments:

Methods: You might want to spell out HOV (human occupied vehicle) for readers who are not familiar with the abbreviation.

Response: this was added to line 374

For the mRNA sampling, it might be worth specifying that tissue samples from individual worms represent biological replicates for the respective condition (and were analyzed separately). Did you take RNA samples from all 30 worm individuals? Cannot hurt to specify it here.

Response: On line 523 this language was added:

“The mRNA was sampled from the trophosome tissue, which contains symbionts, of 30 separate worms. For each condition in the aquarium, three worms were used as biological replicates.”

Results: line 531 “In our DE analyses, nearly all the genes in this cluster all increase or decrease...” remove the second “all” (redundant)

Response: this sentence was removed when editing to shorten manuscript

; line 599: “the NarGHI genes shares” - please correct (the NarGHI genes share)

Response: Good catch, this was corrected

Discussion: line 681: “these genes also have isoforms elsewhere in the genome and could be play an important role” - please correct

Response: We changed it to: “Endoriffia has genes encoding multiple isoforms of PFOR and OGOR, most of which share first neighbors with either the rTCA or the CBB, which may offer a hint at *in vivo* function/direction (see Fig. 5).”

Final Decision Letter:

Message 19th April 2024

:

Dear Pete and Jessie,

I am pleased to accept your Article "Co-expression analysis reveals distinct alliances around two carbon fixation pathways in hydrothermal vent symbionts." for publication in Nature Microbiology. Thank you for having chosen to submit your work to us and many

40congratulations.

Please note that *Nature Microbiology* is a Transformative Journal (TJ). Authors may publish their research with us through the traditional subscription access route or make their paper immediately open access through payment of an article-processing charge (APC). Authors will not be required to make a final decision about access to their article until it has been accepted. Find out more about Transformative Journals

Authors may need to take specific actions to achieve compliance with funder and institutional open access mandates. If your research is supported by a funder that requires immediate open access (e.g. according to Plan S principles) then you should select the gold OA route, and we will direct you to the compliant route where possible. For authors selecting the subscription publication route, the journal's standard licensing terms will need to be accepted, including self-archiving policies. Those licensing terms will supersede any other terms that the author or any third party may assert apply to any

41version of the manuscript.

With kind regards,